# SCALABLE CHAIN OF THOUGHTS VIA ELASTIC REASONING

**Yuhui Xu    Hanze Dong    Lei Wang    Doyen Sahoo    Junnan Li    Caiming Xiong**

Salesforce AI Research

## ABSTRACT

Large reasoning models (LRMs) have achieved remarkable progress on complex tasks by generating extended chains of thought (CoT). However, their uncontrolled output lengths pose significant challenges for real-world deployment, where inference-time budgets on tokens, latency, or compute are strictly constrained. We propose **Elastic Reasoning**, a novel framework for scalable chain of thoughts that explicitly separates reasoning into two phases—*thinking* and *solution*—with independently allocated budgets. At test time, Elastic Reasoning prioritizes the completeness of solution segments, significantly improving reliability under tight resource constraints. To train models that are robust to truncated thinking, we introduce a lightweight *budget-constrained rollout* strategy, integrated into GRPO, which teaches the model to reason adaptively when the thinking process is cut short and generalizes effectively to unseen budget constraints without additional training. Empirical results on mathematical (AIME, MATH500) and programming (LiveCodeBench, Codeforces) benchmarks demonstrate that Elastic Reasoning performs robustly under strict budget constraints, while incurring significantly lower training cost than baseline methods. Remarkably, our approach also produces more concise and efficient reasoning even in unconstrained settings. Elastic Reasoning offers a principled and practical solution to the pressing challenge of controllable reasoning at scale. Our code has been made available at [1].

## 1 INTRODUCTION

Large reasoning models (LRMs) (DeepSeek-AI et al., 2025; OpenAI et al., 2024) have demonstrated remarkable performance on complex reasoning tasks by producing extended Chain-of-Thought (CoT) outputs, which facilitate effective problem-solving in domains such as mathematics and programming. Reinforcement learning (RL) techniques (Schulman et al., 2017; Zelikman et al., 2022; Rafailov et al., 2023; Dong et al., 2023; Shao et al., 2024) have been employed to optimize these reasoning trajectories, enabling LRMs to generate longer, more informative chains. These RL-driven methods scale effectively across diverse benchmarks (Zhang et al., 2024; Dong et al., 2024; Luo et al., 2025c; Xiong et al., 2025b; Luo et al., 2025b), yielding substantial gains in both solution accuracy and robustness; while they often incur significantly longer inference chains (DeepSeek-AI et al., 2025; Du et al., 2025; Yu et al., 2024; Qin et al., 2024; Xiong et al., 2025a). Notably, the length of the reasoning trajectory remains uncontrolled, making it difficult to allocate a fixed compute budget at inference time while maintaining a desired performance level.

Two primary lines of research have been proposed to address this challenge. The first, known as **Long2Short** (Team et al., 2025; Kang et al., 2024), seeks to reduce reasoning length through reinforcement learning with trajectory penalties or compression-aware fine-tuning, where the model is trained on shortened trajectories to preserve performance while minimizing inference cost. The second line of work focuses on **length control** (Muennighoff et al., 2025; Aggarwal & Welleck, 2025; Yuan et al., 2024). S1 (Muennighoff et al., 2025) introduces a simple mechanism that prompts the model to emit special tokens (e.g., "Wait", "Final Answer") to regulate reasoning length. However, this approach significantly degrades performance, as it overlooks the critical role of the solution segment. L1 (Aggarwal & Welleck, 2025) proposes a reinforcement learning framework that enforces

---

[1] https://github.com/SalesforceAIResearch/Elastic-Reasoning

explicit length constraints over the entire trajectory. While more flexible, this method demands substantial training resources and still results in noticeable performance degradation compared to the original model.

We propose **Elastic Reasoning**, a simple yet effective method that enables large reasoning models to achieve scalable and adaptive length control. As illustrated in Figure 1, the S1 approach—generating the answer by emitting a special token such as "Final Answer"—performs better than directly truncating the full reasoning trajectory, underscoring the importance of preserving the solution segment. Motivated by this, we propose separate budgeting which explicitly divides the total token budget $c$ into two parts: $t$ tokens for the *thinking* phase and $s$ tokens for the *solution* phase, where $c = t + s$. Once the model consumes $t$ tokens in the thinking phase, we forcibly terminate it by appending the special token `</think>` and transition to solution generation. Separate budgeting outperforms S1 under varying generation budgets.

To further improve solution quality under incomplete reasoning, we introduce a novel training strategy called *budget-constrained rollout*, which teaches the model to generate high-quality answers even with partial CoT trajectories. This method is integrated into GRPO training and is highly efficient—requiring only 200 training steps on math tasks with a maximum response length of 2K tokens ($t^* = 1K$, $s^* = 1K$), compared to 700 steps for L1-Exact and 820 steps for L1-Max with a 4K response length. Moreover, models trained with Elastic Reasoning generalize effectively to arbitrary reasoning budgets without the need for further fine-tuning.

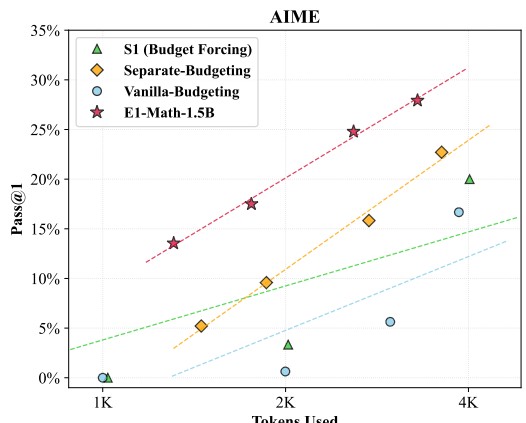

Figure 1: Separating thinking and solution phases allows precise length control. Each point shows Pass@1 at a given token budget, with the x-axis showing average tokens per problem.

We evaluate Elastic Reasoning on both mathematical and programming reasoning tasks, introducing two models: **E1-Math-1.5B** and **E1-Code-14B**. (1) **E1-Math-1.5B** outperforms both L1-Exact and S1, and achieves performance comparable to L1-Max, while requiring significantly fewer training steps. For instance, on the AIME2024 dataset, our method achieves 35.0% accuracy, compared to 27.1% for L1-Max, 24.2% for L1-Exact, and 41.0% for the original model. (2) **E1-Code-14B** demonstrates strong scaling with varying inference budgets, achieving a Codeforces rating of 1987 and placing in the 96.0 percentile—comparable to O1-2024-12-17 (Low), which scores 1991 and ranks in the 96.1 percentile. (3) A surprising observation is that, after training, the trajectories generated by our models are significantly shorter than those from the original DeepScaleR and DeepCoder models across both math and code tasks. This suggests that budget-constrained rollout not only improves length control but also encourages the model to reason more concisely and generate more efficient solutions.

## 2 RELATED WORKS

### 2.1 TEST-TIME SCALING IN LARGE LANGUAGE MODELS

Increasing computation during inference, often referred to as test-time scaling (TTS), has been shown to improve the reasoning capabilities of LLMs (Wei et al., 2023; Wang et al., 2023; Snell et al., 2024; DeepSeek-AI et al., 2025; Team et al., 2025; Muennighoff et al., 2025). Early works, such as chain-of-thought prompting (Wei et al., 2023), show that producing a series of intermediate reasoning steps significantly improves LLMs' performance on complex reasoning tasks. Building on this, self-consistency (Wang et al., 2023) further boosts performance by sampling a diverse set of reasoning paths and selecting the most consistent answer. Recent studies have formalized these findings into test-time inference scaling laws (Snell et al., 2024; Wu et al., 2024). Wu et al. (2024) explore the trade-offs between model size and inference-time computation. Snell et al. (2024) investigated how fixed but non-trivial inference-time budgets can significantly boost LLM performance. The remarkable successes of advanced reasoning models, such as o1 (OpenAI et al., 2024) and

R1 (DeepSeek-AI et al., 2025), have further amplified interest in leveraging TTS techniques. While much of the existing works primarily focuses on improving performance by increasing inference-time computation, our work takes a different perspective: *How can we enable LLMs to perform effective long reasoning under strict output length constraints?*

## 2.2 LENGTH CONTROL IN LARGE LANGUAGE MODELS

Controlling the generation length of an LLM directly affects both latency and monetary cost at inference time. Earlier approaches to length control are designed mainly for general text generation (Jie et al., 2023; Yuan et al., 2024). Typical methods include (i) manipulating positional encodings to achieve exact sequence lengths (Butcher et al., 2024), (ii) modifying training objectives to penalize deviations from length targets (Jie et al., 2023; Singhal et al., 2024), and (iii) fine-tuning on instructions that explicitly state the desired output length (Yuan et al., 2024). Although effective for tasks such as summarization or constrained writing, these techniques generally aim to verbosity or enforce maximum-length limits, and overlook the intricate, step-by-step reasoning processes required for many reasoning tasks. Recent works have begun to explore efficiency in reasoning by encouraging shorter chains (Kang et al., 2024; Arora & Zanette, 2025); however, they typically lack mechanisms for precise, user-defined length targets that align with explicit compute budgets. One notable attempt, budget forcing (Muennighoff et al., 2025), enforces strict token caps by truncating or padding with special tokens. This can yield incomplete reasoning or unnatural, forced outputs, ultimately harming both accuracy and interpretability. Additionally, L1 (Aggarwal & Welleck, 2025) uses reinforcement learning to let models dynamically allocate inference compute based on constraints provided in the prompt. Our approach does not need to include length instructions in the prompt. Instead, we truncate reasoning trajectories to meet a given budget and train the model under these constraints via reinforcement learning.

## 2.3 EFFICIENT REASONING IN LARGE LANGUAGE MODELS

Making complex reasoning in LLMs more efficient, particularly by shortening the reasoning process, is crucial to reducing computational costs and making these models practical for real-world deployment. This has become a vibrant research area with several promising directions to encourage more concise and effective reasoning strategies (Kang et al., 2024; Xu et al., 2024; Hao et al., 2024; Liao et al., 2025; Luo et al., 2025a). One common strategy involves incorporating explicit rewards into RL to encourage the model to find shorter reasoning paths (Team et al., 2025; Luo et al., 2025a). Some focus on creating datasets with examples of concise reasoning paths and then using SFT to teach models how to generate compact and knowledgeable reasoning steps (Kang et al., 2024; Yu et al., 2024). Instead of relying solely on explicit textual reasoning, methods exploring latent reasoning aim to compress these intermediate steps into more compact, internal representations (Hao et al., 2024; Shen et al., 2025; Saunshi et al., 2025). Efficiency can also be improved during inference, without needing to retrain the model. These training-free techniques dynamically adapt the reasoning strategy based on the specific input or task demands (Liao et al., 2025; Fu et al., 2025). In this work, we introduce a training approach using reinforcement learning under strict budget constraints to encourage the model to balance reasoning quality with cost efficiency.

## 3 METHODOLOGY

### 3.1 PRELIMINARIES: REASONING LANGUAGE MODELS

We consider reasoning-augmented language models that generate outputs consisting of two distinct segments: a *thinking* part and a *solution* part. Following prior work, we denote the reasoning phase using special tokens such as `<think>` and `</think>` to explicitly mark the model's intermediate thoughts.

Formally, given an input prompt $x$, the model generates an output sequence $y = (y^{\text{think}}, y^{\text{solution}})$, where $y^{\text{think}}$ contains the intermediate reasoning steps (enclosed between `<think>` and `</think>`) and $y^{\text{solution}}$ contains the final solution. Typically, $y^{\text{think}}$ accounts for most of the total tokens, while $y^{\text{solution}}$ provides a concise summary and final answer. The overall generation structure is:

$$y = (\texttt{<think>} \text{ intermediate reasoning } \texttt{</think>}, \text{ solution})$$

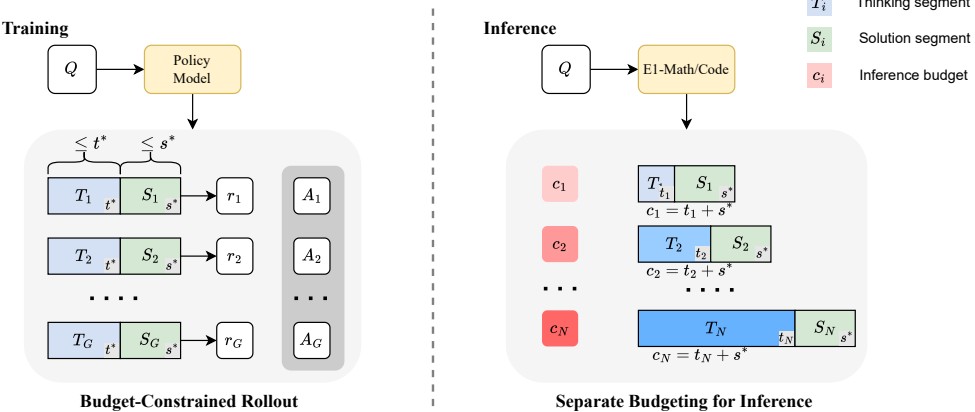

Figure 2: The framework of Elastic Reasoning. Elastic Reasoning comprises two key components: (1) GRPO training with budget-constrained rollout, and (2) separate budgeting for inference. **Left:** During training, the model is optimized using GRPO under a fixed token budget $(t^*, s^*)$. **Right:** At inference time, the trained E1 model can generalize to arbitrary token budgets $c_i = t_i + s^*$, enabling flexible and efficient reasoning. The red squares are visual markers for different budget settings.

## 3.2 ELASTIC REASONING

### 3.2.1 BUDGET-CONSTRAINED INFERENCE

In many real-world applications, inference cost must be carefully controlled due to constraints on latency, computation, or memory. A common approach is to truncate generation after a fixed number of tokens $c$, enforcing:

$$|y| \leq c$$

where $|y|$ denotes the number of generated tokens. However, naively truncating the output often results in incomplete or missing $y^{\text{solution}}$, leading to invalid or unusable predictions.

### 3.2.2 SEPARATE BUDGETING FOR THINKING AND SOLUTION

To address this limitation, we propose *Separate Budgeting*, a method that explicitly allocates independent budgets for the reasoning and solution phases. A key observation is that even when the reasoning phase is forcibly terminated (e.g., by inserting `</think>`), the model is still capable of producing a coherent—and often correct—solution.

Given a total generation budget $c$, we divide it into two components: a budget $t$ for the thinking phase and a budget $s$ for the solution phase, such that $c = t + s$.

During inference:

- The model begins generating within a `<think>` block.
- If the model emits `</think>` before reaching the budget $t$, we transition immediately to the solution phase.
- If the budget $t$ is exhausted before `</think>` is emitted, we forcibly terminate the reasoning by appending `</think>`.
- The model then continues generating the solution segment, up to a maximum of $s$ tokens.

This approach ensures that both the reasoning and solution components are explicitly accommodated within the total budget $c$, thereby avoiding unintended truncation of the solution segment. The thinking budget $t$ can be flexibly adjusted at inference time to match different application scenarios, while the solution phase always retains a guaranteed allocation. As shown in Figure 1, Separate Budgeting outperforms both vanilla budgeting (naïve truncation) and S1 (budget forcing). By ded-

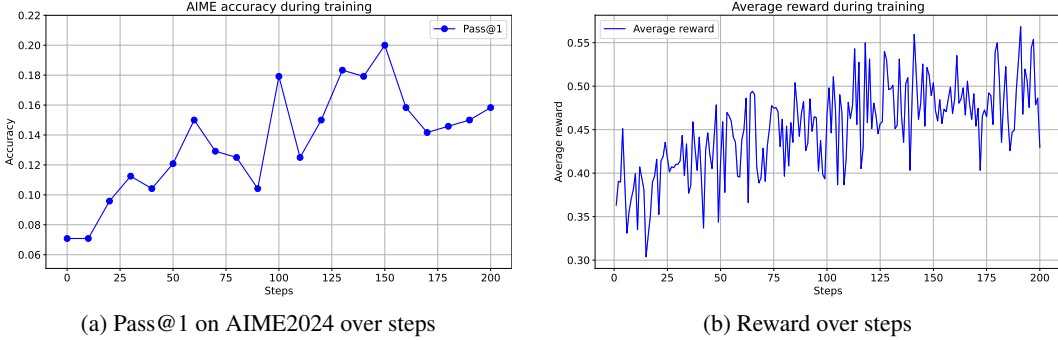

(a) Pass@1 on AIME2024 over steps

(b) Reward over steps

Figure 3: Validation accuracy and reward curves of E1-Math-1.5B over training steps.

icating a fixed token budget for solution generation, Separate Budgeting significantly improves the reliability and quality of model outputs under tight inference-time constraints.

### 3.2.3 BUDGET-CONSTRAINED ROLLOUT

While Separate Budgeting ensures dedicated budgets for both reasoning and solution phases, we observe that naively truncating the thinking part—especially on complex tasks such as code generation—can lead to significant performance degradation. To mitigate this issue, we propose a reinforcement learning (RL) fine-tuning procedure that explicitly trains the model under reasoning budget constraints, allowing it to produce more effective and concise reasoning within limited budgets.

We adopt GRPO as our RL algorithm. Let $\pi_\theta$ denote the policy of a language model parameterized by $\theta$, which generates a response $y = (y^{\text{think}}, y^{\text{solution}})$ for a given input $x$, subject to a total budget constraint $t^* + s^* = c^*$. In standard GRPO, the policy generates a full trajectory without any structural constraints. In contrast, our budget-constrained rollout conditions the policy on a fixed thinking/solution budget pair $t^*, s^*$. During training, we simulate the Separate Budgeting procedure used at inference time: the policy rolls out a reasoning segment $y^{\text{think}}$ up to a maximum of $t^*$ tokens. If the model emits the `</think>` token before reaching this limit, it proceeds to generate the solution segment as usual. Otherwise, we forcibly append `</think>` once the budget $t^*$ is reached. The model then generates the solution segment $y^{\text{solution}}$ using the remaining $s^*$ tokens.

Let $r(y)$ denote a task-specific reward function. The training objective is to maximize the expected reward:

$$J(\theta) = \mathbb{E}_{x\sim\mathcal{D},\ y\sim\pi_\theta(\cdot|x;\ t^*,s^*)}\left[r(y)\right]$$

We optimize $J(\theta)$ using GRPO with the following gradient estimator:

$$\nabla_\theta J(\theta) = \mathbb{E}_{x,y}\left[A(x,y)\nabla_\theta \log \pi_\theta(y \mid x;\ t^*,s^*)\right], \quad A(x,y) = \frac{r(y) - \mathbb{E}_{y'\sim\pi_\theta(\cdot|x;\ t^*,s^*)}\left[r(y')\right]}{\sqrt{\mathbb{V}_{y'\sim\pi_\theta(\cdot|x)}\left[r(y')\right]}}$$

where the conditioning in $\pi_\theta(\cdot \mid x;\ t^*,s^*)$ simply indicates that trajectories are sampled under these rollout constraints, rather than from unconstrained generation. In our training setup, we fix the budget pair to $(t^*, s^*) = (1\text{K}, 1\text{K})$ for simplicity and efficiency. Surprisingly, we find that the learned policy generalizes well to a wide range of unseen budget configurations at test time, without requiring any additional fine-tuning. As shown in Figure 1, the E1-Math-1.5B model achieves substantial improvements while generalizing robustly across various generation budgets. This indicates that Elastic Reasoning encourages the model to internalize a flexible reasoning strategy that adapts to different resource constraints. This RL-based adaptation helps the model prioritize informative reasoning content earlier in the generation process, thereby improving both robustness and solution quality under test-time truncation.

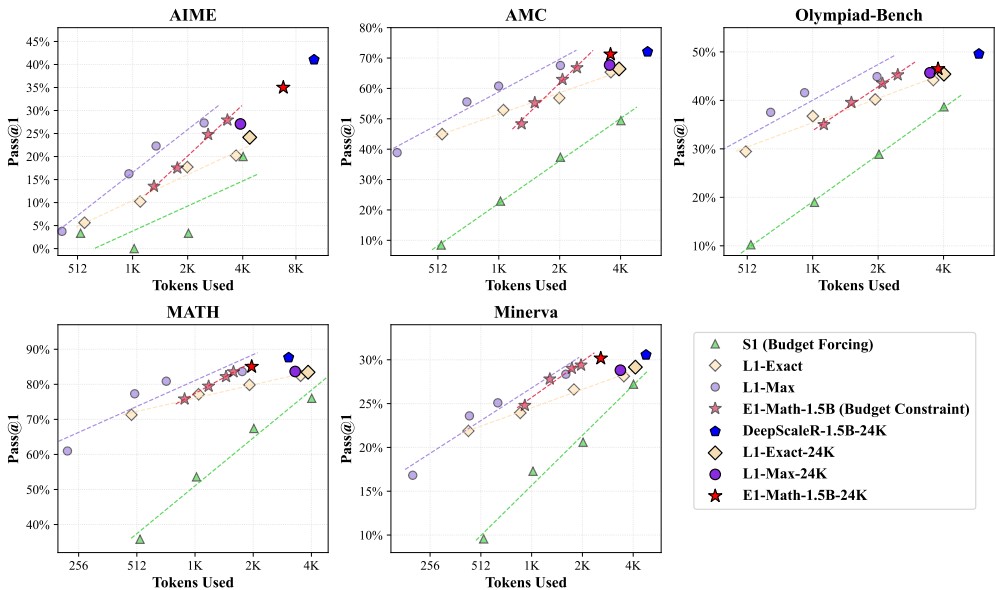

Figure 4: Comparison of E1-Math-1.5B with L1 and S1 baselines under varying generation budgets. Each point shows Pass@1 at a given token budget, with the x-axis showing average tokens per problem. 24K / 64K settings correspond to the maximum token lengths permitted during inference, with no explicit budget constraints imposed.

# 4 EXPERIMENT RESULTS

## 4.1 MODELS AND DATASETS

Our base models are DeepScaleR-1.5B-Preview (Luo et al., 2025c) and DeepCoder-14B-Preview (Luo et al., 2025b), which are fine-tuned from DeepSeekR1-Distill-Qwen-1.5B and 14B (DeepSeek-AI et al., 2025) through iterative context lengthening. For training data, we follow the same datasets used in (Luo et al., 2025c;b). In the math domain, the training set consists of AIME (1984-2023), AMC, Omni-Math (Gao et al., 2024), and STILL (Min et al., 2024). For code training, we use TACO (Li et al., 2023), SYNTHETIC-1 (Mattern et al., 2025), and LiveCodeBench (2023/05/01-2024/07/31) (Jain et al., 2024). For evaluation, we use AIME 2024, MATH500 (Hendrycks et al., 2021), AMC, Olympiad-Bench (Gao et al., 2024), and Minerva Math (Lewkowycz et al., 2022) for mathematical reasoning. For code-related tasks, we evaluate on LiveCodeBench (2024/08/01-2025/02/01) (Jain et al., 2024), Codeforces, and HumanEval+ (Liu et al., 2023). For mathematical reasoning tasks, we report averages over 16 runs, whereas for code-related tasks the results are averaged over 8 runs. 24K / 64K settings correspond to the maximum token lengths permitted during inference, with no explicit budget constraints imposed. More training details are in Appendix B.

## 4.2 MATHEMATICAL REASONING RESULTS

We visualize the reward and validation Pass@1 performance on AIME2024 every 10 steps during training in Figure 3. It can be observed that the reward steadily increases during the initial training phase and begins to converge after approximately the 150[th] step. Meanwhile, the validation accuracy (Pass@1) improves rapidly, rising from around 0.07 to 0.20 over the course of training. This demonstrates that, through budget-constrained rollout, the model can quickly learn to reason effectively when the thinking phase is incomplete.

We report Pass@1 accuracy versus the number of tokens used across five math benchmarks: AIME, AMC, Olympiad-Bench, MATH500, and Minerva Math in Figure 4. Our proposed method, E1-Math-1.5B, under both budget-constrained and 24K-token settings (red stars), consistently outperforms S1 (Budget Forcing) and L1-Exact, and performs competitively with L1-Max, while requiring significantly fewer training steps. On MATH500, E1-Math-1.5B achieves a Pass@1 accuracy

Table 1: Comparison of models across LiveCodeBench, Codeforces, HumanEval+, and AIME benchmarks. E1-code-14B variants trained exclusively on code data; their AIME scores, obtained on math problems unseen during training, demonstrate that E1-code-14B retains strong math performance.

| Model | LiveCodeBench | Codeforces Rating | Codeforces Percentile | HumanEval+ | AIME |
|---|---|---|---|---|---|
| O1-2024-12-17 (Low) | 59.5 | **1991** | **96.1** | 90.8 | **74.4** |
| O3-Mini-2025-1-31 (Low) | 60.9 | 1918 | 94.9 | **92.6** | 60.0 |
| O1-Preview | 42.7 | 1658 | 88.5 | 89.0 | 40.0 |
| DeepSeek-R1 | **62.8** | 1948 | 95.4 | **92.6** | **79.8** |
| DeepSeek-R1-Distill-Qwen-14B | 53.0 | 1791 | 92.7 | 92.0 | 69.7 |
| DeepCoder-14B-Preview[2] | 58.1 | 1945 | 95.4 | 90.8 | 71.7 |
| E1-code-14B ($t = 1k, a = 1k$) | 37.3 | 1457 | 78.1 | 88.3 | 17.9 |
| E1-code-14B ($t = 2k, a = 1k$) | 41.6 | 1604 | 85.4 | 89.6 | 28.5 |
| E1-code-14B ($t = 3k, a = 1k$) | 44.1 | 1711 | 90.6 | 90.8 | 35.4 |
| E1-code-14B ($t = 4k, a = 1k$) | 47.0 | 1771 | 92.3 | 92.0 | 41.9 |
| E1-code-14B | $58.4_{+0.3}$ | $1987_{+42}$ | $96.0_{+0.6}$ | $91.4_{+0.6}$ | 70.6 |

of 83.6% using only 1619 tokens per question, whereas L1-Exact and L1-Max yield lower or comparable performance with more tokens (L1-Exact: 79.9% with 1959 tokens; L1-Max: 83.6% with 1796 tokens). Notably, when evaluated without inference-time budget constraints, E1-Math-1.5B achieves higher accuracy than all baseline methods across all benchmarks. For example, on AIME2024, E1-Math-1.5B exhibits a performance degradation of only 6.0% relative to the original model, compared to 12.9% for L1-Max and 16.8% for L1-Exact. These results demonstrate that our method is not only effective in enforcing inference-time budget constraints but also preserves most of the original model's performance. When compared with the original DeepScaleR-1.5B, E1-Math-1.5B reduces the average number of tokens used across datasets by more than 30%, including a 32.1% reduction on AIME2024 (see further analysis in Appendix C).

Furthermore, similar to L1, S1, and O1, we observe a clear log-linear scaling pattern in E1: performance improves approximately linearly with respect to the logarithm of the number of generated reasoning tokens. We provide additional results for other model variants in Appendix I.

## 4.3 CODE REASONING RESULTS

As shown in Figure 5, we visualize the Pass@1 accuracy on LiveCodeBench under varying generation budgets, comparing our method to a simple separate budgeting strategy for thinking and solution. We observe that the original **DeepCoder-14B-Preview** fails to generate correct outputs when reasoning is incomplete, consistently achieving less than 10% accuracy when inference budget is less than 4K even using separate budgeting. In contrast, our E1-Code-14B model demonstrates impressive scalability: its performance improves steadily as the inference budget increases, highlighting the effectiveness of our training strategy in enabling the model to reason adaptively under constrained thinking. Notably, E1-Code-14B also achieves a performance improvement of 0.3% on LiveCodeBench even in the unconstrained setting, while simultaneously reducing the average number of generated tokens by

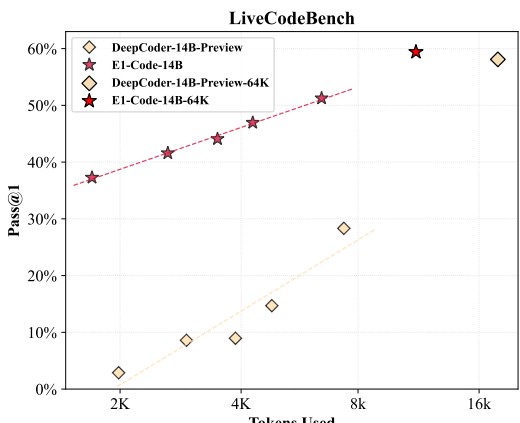

Figure 5: Pass@1 accuracy on LiveCodeBench under varying reasoning budgets with the x-axis showing average tokens per problem. Both of the models inference with separate budgeting.

**37.4%**—from 17,815 to 11,145 tokens. This indicates that our method not only scales well with inference budgets but also promotes more concise and efficient reasoning.

---

[2]Results are reproduced using the authors' official code and model with the same evaluation protocol.

In Table 1, we report the results of E1-Code-14B on four benchmarks: LiveCodeBench, Codeforces, HumanEval Plus, and AIME2024. We observe consistent test-time scaling behavior across all benchmarks under constrained inference budgets. Beyond scalability, our model also demonstrates strong performance in the unconstrained setting. Specifically, we observe performance improvements on LiveCodeBench, Codeforces, and HumanEval Plus, and only a slight performance drop on AIME2024. On Codeforces, E1-Code-14B achieves a 42-point improvement in rating and a 0.6 percentile gain, outperforming O3-Mini-2025-1-31 (Low) and performing comparably to O1-2024-12-17 (Low). These results highlight that our method not only enables efficient, budget-constrained reasoning but also enhances overall reasoning capability, even in unconstrained scenarios.

## 4.4 ANALYSIS AND DISCUSSIONS

In this section, we provide ablation studies examining which components of the model are enhanced after training, the effect of the training budget $t^*$, and the allocation of tokens between the thinking and solution phases. Additional ablations on varying the solution-token budget and the zero-thinking setting are provided in Appendix J and Appendix K.

### 4.4.1 WHICH PART IS ENHANCED AFTER TRAINING?

Table 2: Ablation of enhanced thinking and solution on DeepScaleR-1.5B-Preview and E1-Math-1.5B. Budget is in format 'thinking+solution' (in thousands of tokens).

| DeepScaleR-1.5B | | E1-Math-1.5B | | Pass@1 (%) | | | |
|---|---|---|---|---|---|---|---|
| Thinking | Solution | Thinking | Solution | 0.5K+1K | 1K+1K | 2K+1K | 3K+1K |
| ✓ | ✓ | | | 2.10 | 4.80 | 12.5 | 20.0 |
| | ✓ | ✓ | | $3.50_{+1.4}$ | $7.90_{+3.1}$ | $20.6_{+8.1}$ | $24.0_{+4.0}$ |
| ✓ | | | ✓ | $10.8_{+8.7}$ | $14.2_{+9.4}$ | $21.9_{+9.4}$ | $26.4_{+6.4}$ |
| | | ✓ | ✓ | $13.5_{+11.4}$ | $17.5_{+12.7}$ | $24.8_{+12.3}$ | $27.9_{+7.9}$ |

To better understand which components of the reasoning process are enhanced through training, we conduct ablation experiments on DeepScaleR-1.5B-Preview and E1-Math-1.5B using the AIME2024 benchmark. Specifically, we separately generate the *thinking* and *solution* segments using both models under varying generation budgets. For example, we use DeepScaleR-1.5B-Preview to generate the thinking part, and then use E1-Math-1.5B to generate the corresponding solution based on that reasoning. This setup allows us to isolate the contributions of each model to the reasoning pipeline and assess how training improves each component.

As shown in Table 2, we observe that both the thinking and solution are enhanced after training. Notably, the improvement in the solution component is more substantial, particularly under constrained thinking budgets. For instance, using the E1 model to generate only the solution segment yields an 8.7% gain in accuracy compared to using the original DeepScaleR model, under a generation budget of $(0.5\text{K}+1\text{K})$ tokens. This highlights the effectiveness of budget-constrained rollout in strengthening the model's ability to produce high-quality solutions based on incomplete reasoning.

This observation also helps explain why training with a fixed budget constraint (e.g., $(1\text{K}, 1\text{K})$) enables the model to generalize effectively to a wide range of budget configurations. We hypothesize that the improvement in solution generation plays a central role in this generalization, allowing the model to adapt even when the available thinking tokens are reduced.

### 4.4.2 ABLATION OF TRAINING BUDGET $t^*$

To further investigate the role of the thinking budget $t^*$ in our proposed budget-constrained rollout, we conduct experiments to evaluate the model's performance under four settings: $t^* \in \{0.5\text{K}, 1\text{K}, 2\text{K}, 3\text{K}\}$, while keeping the solution budget fixed at $a^* = 1\text{K}$. We evaluate on five math benchmarks: AIME, AMC, Olympiad-Bench, MATH500, and Minerva Math (Figure 6).

Across all configurations, the model demonstrates strong generalization to varying inference budgets on all benchmarks. Among the tested values, $t^* = 1\text{K}$ consistently achieves the best performance, while also maintaining a low maximum generation length of 2K tokens, making it a highly efficient

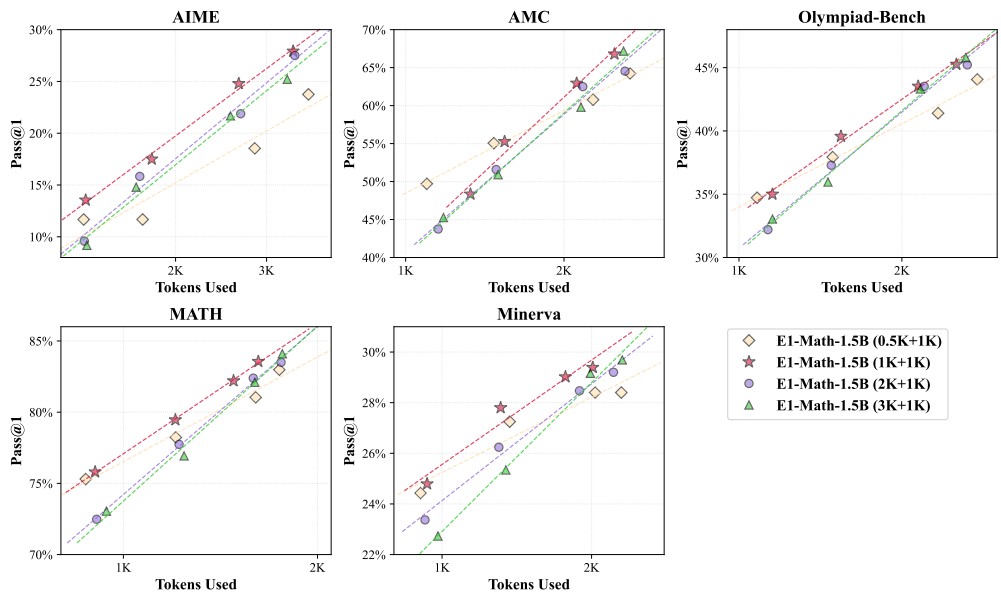

Figure 6: Ablation study on training reasoning budget $t^*$. We compare four settings: $t^* \in \{0.5K, 1K, 2K, 3K\}$, while keeping the solution budget fixed at $a^* = 1K$. Each point shows Pass@1 at a given token budget, with the x-axis showing average tokens per problem.

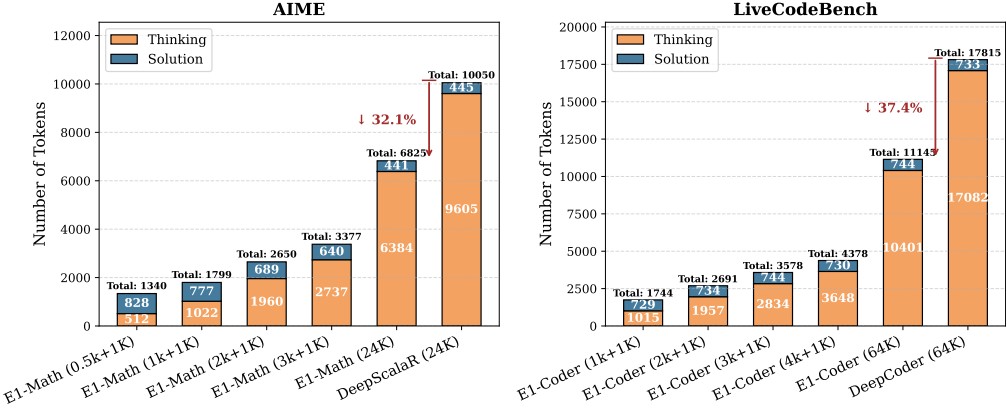

Figure 7: Distribution of tokens for thinking and solution across different generation budgets.

and effective setting. Based on the trade-off between performance and computational cost, we adopt $(t^* = 1K, \ s^* = 1K)$ as our default configuration (Appendix H).

### 4.4.3 TOKEN ALLOCATION BETWEEN THINKING AND SOLUTION

Figure 7 visualizes the distribution of *thinking* and *solution* tokens within generated trajectories under different generation budget constraints. We select AIME2024 for the math task and Live-CodeBench for the coding task.

For AIME2024, as the inference budget decreases, the number of tokens used in the thinking segment decreases accordingly, while the number of tokens in the solution segment slightly increases. A similar trend is observed on LiveCodeBench, where the thinking tokens decrease with tighter budgets, while the number of solution tokens remains relatively stable.

Notably, even when evaluated without budget constraints, our trained E1 models demonstrate substantial token efficiency: they reduce total token usage by 32.1% on AIME2024 and 37.4% on Live-CodeBench, while maintaining strong performance (even slightly better than the baseline model). This suggests that the model has learned to reason more concisely and generate efficient solutions post training. Qualitative analysis is provided in Appendix G.

## 5 CONCLUSION

We introduce **Elastic Reasoning**, a unified framework for enabling large reasoning models to generate accurate and efficient chain-of-thought outputs under strict inference-time constraints. By explicitly separating the reasoning process into *thinking* and *solution* phases, and training with a novel *budget-constrained rollout* strategy, our approach ensures robustness to truncated reasoning while preserving or even improving overall performance. Elastic Reasoning significantly reduces token usage during inference, generalizes across unseen budget configurations, and outperforms prior length control baselines in both mathematical and programming domains. Our findings offer a scalable and principled solution for real-world deployment of reasoning LLMs where computation budgets are limited. We believe this framework opens new directions for budget-aware reasoning.

## 6 ETHICS STATEMENT

Our study focuses on improving the efficiency and scalability of reasoning in large language models by introducing budget-constrained training and inference. All experiments were conducted on publicly available datasets in mathematics (e.g., AIME, AMC, MATH500, Minerva Math) and programming (e.g., LiveCodeBench, Codeforces, HumanEval+), without the use of private or sensitive user data. No personally identifiable information (PII) or human subject data was collected, and therefore IRB approval was not required.

We acknowledge the broader ethical considerations surrounding large language models, including potential risks of misuse, fairness, and environmental impact. Our method, Elastic Reasoning, is designed to reduce inference costs by making models more token-efficient. This can mitigate environmental impact by lowering computational and energy requirements.

## 7 REPRODUCIBILITY STATEMENT

We have taken several steps to facilitate reproducibility of our work. An anonymous implementation of our proposed Elastic Reasoning framework is included in the supplementary materials. The main text and Appendix B describe the methodology and training procedure in detail, including the budget-constrained rollout strategy and evaluation protocol. We provide complete information about the datasets used in both the math and code domains (Section 4.1). Hyperparameters, training configurations, and additional ablation studies are documented in the Appendix to enable faithful reproduction of our results. Together, these resources ensure that both the reported results and future extensions of this work can be reproduced by the community.

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

## A   USE OF LARGE LANGUAGE MODELS (LLMS)

We used large language models (LLMs) solely as a writing assist tool to improve grammar, clarity, and style. The research ideas, methodology, experiments, analysis, and conclusions were developed entirely by the authors without reliance on LLMs. No part of the scientific contribution, experimental design, or result interpretation involved LLM usage.

## B   TRAINING DETAILS

For GRPO training, we adopt the same hyperparameters as those used in DeepScaleR-1.5B-Preview and DeepCode-14B-Preview. For E1-Math-1.5B, we use a learning rate of $1 \times 10^{-6}$ and a batch size of 128. The maximum context length is set to 1K tokens for the thinking segment and 1K tokens for the solution segment. Training is performed for 200 steps using the `VeRL` (Sheng et al., 2024) framework. For E1-Code-14B, we use the same learning rate and batch size. The context length configuration mirrors that of the math model: 1K tokens for thinking and 1K tokens for solution. Training is conducted for only 30 steps.

| Model | AMC | Olympiad-Bench | MATH500 | Minerva Math |
|---|---|---|---|---|
| DeepScaleR-1.5B-24K | 72.06 (5571) | 49.59 (5956) | 87.63 (3124) | 30.56 (4881) |
| L1-Exact-24K | 66.42 (4024) | 45.41 (4111) | 83.45 (3937) | 29.16 (4221) |
| L1-Max-24K | 67.70 (3615) | 45.69 (3547) | 83.66 (3375) | 28.81 (3440) |
| E1-Math-1.5B-24K | 71.23 (3653) | 46.50 (3865) | 85.05 (2011) | 30.17 (2631) |

Table C.1: Pass@1 accuracy (%) under 24K-token unconstrained inference across math benchmarks. Numbers in parentheses indicate average tokens used.

**Reward design.** We follow the reward function used in **DeepScaleR-1.5B**, which is standard in recent Long-CoT RL works. The reward is binary:

- **1** — if the final answer is correct according to our automatic LaTeX/Sympy checker.

- **0** — otherwise (incorrect answer, malformed output, or missing `<think>` / `</think>` delimiters).

Importantly, since our rollout is **budget-constrained**, the reward is computed based on the trajectory generated under the 1K+1K budget. This ensures that the reward signal is aligned with the intended inference-time behavior.

**Hyperparameter settings.** We follow **exactly the same hyperparameters** as DeepScaleR-1.5B for fair comparison, except for our modified rollout procedure:

- Batch size: **128**

- Learning rate: **1e-6**

- Rollout sampling temperature: **0.6**

- Number of rollouts per prompt: **8**

- Budget-constrained rollout: **1K** thinking tokens, **1K** solution tokens

- Total RL steps: **200** (math), **30** (code)

## C  COMPARISON UNDER 24K-TOKEN UNCONSTRAINED INFERENCE

To highlight the effectiveness of our method, we compare E1-Math-1.5B with L1-Exact, L1-Max, and the vanilla DeepScaleR-1.5B under the 24K-token unconstrained inference setting across five math benchmarks (Table C.1). Note that E1-Math-1.5B, L1-Exact, and L1-Max are all fine-tuned from DeepScaleR-1.5B.

Compared to L1-Exact and L1-Max, our method achieves the best overall trade-off between performance and length control. For example, on Minerva Math, E1-Math-1.5B uses only 2,631 tokens on average (vs. 3,440 for L1-Max and 4,221 for L1-Exact), while achieving 30.17% accuracy—1.36% higher than L1-Max and 1.01% higher than L1-Exact.

Relative to the vanilla DeepScaleR-1.5B, our accuracy remains comparable while requiring substantially fewer tokens. For instance, on AMC, E1-Math-1.5B achieves 71.23% accuracy with 3,653 tokens, whereas DeepScaleR-1.5B requires 5,571 tokens to reach 72.06%. These results demonstrate that our approach preserves accuracy while significantly improving length efficiency.

## D  REASONING COMPLETENESS RATIO

We measure the **reasoning completeness ratio**—the fraction of cases where the full reasoning fits within the thinking budget—and the corresponding Pass@1 accuracy under different budget settings on AIME, averaged over 16 runs.

| Inference Budget | 0.5K+1K | 1K+1K | 2K+1K | 3K+1K |
|---|---|---|---|---|
| Complete Ratio (vanilla) | 0.0% | 0.0% | 2.3% | 5.4% |
| Accuracy (vanilla) | 5.2% | 9.6% | 15.8% | 22.7% |
| Complete Ratio (E1) | 0.0% | 1.3% | 7.5% | 9.2% |
| Accuracy (E1) | 13.5% | 17.5% | 24.8% | 27.9% |

Table D.1: Reasoning completeness ratio and Pass@1 accuracy on AIME.

At low budgets (0.5K+1K and 1K+1K), the completeness ratio remains near zero, even after training. Nevertheless, E1 achieves substantially higher accuracy, indicating that it learns to be robust to truncated reasoning. E1 becomes more token-efficient over time, which slightly increases the completeness ratio as a byproduct—but this is not the main source of its gains.

Additional evidence comes from Table 2: when comparing the first and third rows, the thinking segment is held fixed, so the completeness ratio remains the same. Yet, E1 significantly outperforms the vanilla model in accuracy. This confirms that E1's improvements arise not from completing reasoning more often, but from **better use of limited budgets**.

## E  ITERATIVE TRAINING

Table E.1: Pass@1(%) on AIME 2024 across different budget configurations in two iterations.

| Iteration | 0.5K+1K | 1K+1K | 2K+1K | 3K+1K |
|---|---|---|---|---|
| 1$^{st}$ (1K+1K) | 13.5 | 17.5 | 24.8 | 27.9 |
| 2$^{nd}$ (3K+1K) | 11.5 | 17.1 | 22.9 | 26.7 |
| 1$^{st}$ (3K+1K) | 9.2 | 14.8 | 21.7 | 25.2 |
| 2$^{nd}$ (1K+1K) | 11.7 | 15.0 | 22.9 | 25.5 |

We investigate whether the model benefits from *iterative training*, i.e., performing a second round of training with a different compute budget. Concretely, we consider two schedules: (i) train with a budget of ($t^* = 1K$, $a^* = 1K$) and continue from the checkpoint with ($t^* = 3K$, $a^* = 1K$); and (ii) the reverse order. Results on AIME2024 (Table E.1) show no noticeable improvement from the second round. This suggests that once the model has learned to reason effectively under a given budget, further training with a different budget provides little additional benefit.

## F  MMLU RESULTS

To further assess generalization, we evaluate both models on MMLU (Hendrycks et al., 2020). We adopt the same budget-constrained settings and additionally report results under unconstrained inference (24K). All results are averaged over 16 runs:

| Inference Budget | 0.5K+1K | 1K+1K | 2K+1K | 3K+1K | 24K |
|---|---|---|---|---|---|
| Base Model | – | – | – | – | 48.3% |
| 1st Iteration (1K+1K) | 47.4% | 47.7% | 48.1% | 48.6% | 48.4% |
| 2nd Iteration (3K+1K) | 47.4% | 47.6% | 47.9% | 48.2% | 48.2% |

Table F.1: Performance on MMLU under different inference budgets, averaged over 16 runs.

The second-iteration model exhibits no degradation in general capability. This demonstrates that E1 retains its broad reasoning ability even after adapting to a larger budget, confirming that it learns *budget handling as a transferable skill* without forgetting prior competencies.

## G QUALITATIVE ANALYSIS

To examine how reasoning adapts under budget constraints, we compare outputs from the original model and our E1 model (trained with a 0.5K thinking + 1K solution budget) on an AIME2024 example.

**Original model (1537 tokens).** The reasoning path is verbose, with many tokens consumed within individual steps.

**E1 model (1091 tokens).** After training with a constrained budget, the reasoning steps become denser and more structured, reflecting more purposeful use of limited space. The model transitions more smoothly and directly into the solution, while maintaining accuracy. The final answer is reached correctly and with fewer tokens.

```
<think>
Okay, so I have this problem about Aya's walk. Let me try to understand
    ↪ what it's asking.

Aya walks 9 kilometers every morning. When she walks at a speed of s
    ↪ kilometers per hour, it takes her 4 hours, including t minutes
    ↪ spent in the coffee shop. Then, when she increases her speed to s+2
    ↪  km/h, the total time becomes 2 hours and 24 minutes, including the
    ↪  same t minutes in the coffee shop. Now, we need to find the time
    ↪ it takes if she walks at s + 1/2 km/h, again including t minutes in
    ↪  the coffee shop.

Alright, let's break this down step by step.

First, when she walks at speed s km/h, the total time including the
    ↪ coffee shop is 4 hours. That total time includes both walking and
    ↪ the time in the coffee shop. Similarly, when her speed increases to
    ↪  s+2 km/h, the total time becomes 2 hours and 24 minutes, which is
    ↪ 2 + 24/60 = 2 + 0.4 = 2.4 hours.

So, let's denote:

- Let t be the time spent in the coffee shop in minutes.
- Let d be the distance she walks before entering the coffee shop. Wait,
    ↪ but actually, the total distance is 9 km. So, perhaps it's better
    ↪ to consider the walking time and the coffee shop time separately.

Wait, perhaps another approach: the total time is walking time plus
    ↪ coffee shop time. So, when she walks at s km/h, the total time is 4
    ↪  hours, which is walking time + t minutes. But t is in minutes, so
    ↪ perhaps we need to convert everything to hours or minutes.

Let me clarify the units.

The walking speed is in km/h, so the walking time would be distance/speed
    ↪ . The coffee shop time is in minutes, so we need to convert it to
    ↪ hours or keep it as minutes. It's probably easier to convert
    ↪ everything to hours.

So, let's define:

- When she walks at speed s km/h, total time is 4 hours, which includes t
    ↪  minutes in the coffee shop. So, the walking time is 4 - (t/60)
    ↪ hours.

Similarly, when she walks at s+2 km/h, the total time is 2 hours and 24
    ↪ minutes, which is 2 + 24/60 = 2 + 0.4 = 2.4 hours</think>

Let's denote:
```

```
- \( s \) as her walking speed in km/h.
- \( t \) as the time spent in the coffee shop in minutes.
- The total time when walking at \( s \) km/h is 4 hours, including \( t
    ↪ \) minutes in the coffee shop.

Therefore, the walking time is \( 4 - \frac{t}{60} \) hours.

Since she walks 9 km, the walking time can also be expressed as \( \frac
    ↪ {9}{s} \) hours.

So, we have:
\[ \frac{9}{s} = 4 - \frac{t}{60} \quad \text{(1)} \]

Similarly, when she walks at \( s + 2 \) km/h, the total time is 2 hours
    ↪ and 24 minutes, which is \( 2 + \frac{24}{60} = 2.4 \) hours.

Again, the walking time is \( 2.4 - \frac{t}{60} \) hours.

Expressing walking time as \( \frac{9}{s + 2} \) hours, we get:
\[ \frac{9}{s + 2} = 2.4 - \frac{t}{60} \quad \text{(2)} \]

Now, we have two equations:
1. \( \frac{9}{s} = 4 - \frac{t}{60} \)
2. \( \frac{9}{s + 2} = 2.4 - \frac{t}{60} \)

Let me denote \( \frac{t}{60} \) as \( x \) for simplicity. Then,
    ↪ equation (1) becomes:
\[ \frac{9}{s} = 4 - x \quad \text{(1a)} \]

Equation (2) becomes:
\[ \frac{9}{s + 2} = 2.4 - x \quad \text{(2a)} \]

From equation (1a), we can express \( x \) as:
\[ x = 4 - \frac{9}{s} \quad \text{(1b)} \]

Substitute equation (1b) into equation (2a):
\[ \frac{9}{s + 2} = 2.4 - \left(4 - \frac{9}{s}\right) \]
Simplify:
\[ \frac{9}{s + 2} = 2.4 - 4 + \frac{9}{s} \]
\[ \frac{9}{s + 2} = -1.6 + \frac{9}{s} \]
Bring all terms to one side:
\[ \frac{9}{s + 2} - \frac{9}{s} = -1.6 \]
Factor out 9:
\[ 9\left( \frac{1}{s + 2} - \frac{1}{s} \right) = -1.6 \]
Simplify the expression inside the parentheses:
\[ \frac{1}{s + 2} - \frac{1}{s} = \frac{s - (s + 2)}{s(s + 2)} = \frac
    ↪ {-2}{s(s + 2)} \]
Thus:
\[ 9 \times \left( \frac{-2}{s(s + 2)} \right) = -1.6 \]
Simplify:
\[ \frac{-18}{s(s + 2)} = -1.6 \]
Multiply both sides by -1:
\[ \frac{18}{s(s + 2)} = 1.6 \]
Solve for \( s(s + 2) \):
\[ s(s + 2) = \frac{18}{1.6} = \frac{180}{16} = \frac{45}{4} = 11.25 \]
So:
\[ s^2 + 2s - 11.25 = 0 \]
Solve this quadratic equation:
Using quadratic formula:
\[ s = \frac{ -2 \pm \sqrt{4 + 45} }{2} = \frac{ -2 \pm \sqrt{49} }{2} =
    ↪ \frac{ -2 \pm 7 }{2} \]
We discard the negative solution since speed can't be negative:
\[ s = \frac{5}{2} = 2.5 \text{ km/h} \]

Now, substitute \( s = 2.5 \) back into equation (1a):
```

```
\[ \frac{9}{2.5} = 4 - x \]
\[ 3.6 = 4 - x \]
Thus:
\[ x = 4 - 3.
```

Listing 1: Response from original model with 1537 tokens.

```
<think>
Alright, so I've got this problem about Aya's walk and coffee shop time.
    ↪ Let me try to break it down step by step.

First, the problem says Aya walks 9 kilometers every morning. When she
    ↪ walks at a speed of s km/h, the total time taken, including the
    ↪ coffee shop time, is 4 hours. Then, when her speed increases to s+2
    ↪  km/h, the total time becomes 2 hours and 24 minutes. In both cases
    ↪ , the time spent in the coffee shop is the same, denoted as t
    ↪ minutes.

The question is asking, if Aya walks at s + 0.5 km/h, how many minutes
    ↪ will her total walk time be, including the coffee shop time.

Okay, let's parse this information.

First scenario:
- Speed: s km/h
- Distance: 9 km
- Total time including coffee: 4 hours
- Therefore, time spent walking + time in coffee = 4 hours

Second scenario:
- Speed: s + 2 km/h
- Distance: 9 km
- Total time including coffee: 2 hours 24 minutes, which is 2 + 24/60 = 2
    ↪  + 0.4 = 2.4 hours

Third scenario:
- Speed: s + 0.5 km/h
- Distance: 9 km
- Total time: let's denote as T hours, which includes t minutes in the
    ↪ coffee shop.

We need to find T in hours, then convert it to minutes.

Let's start by setting up equations.

In the first case, the time spent walking is distance divided by speed,
    ↪ so 9/s hours. Then, the time in the coffee shop is t minutes, which
    ↪  is t/60 hours. Therefore, total time is 9/s + t/60 = 4 hours.

In the second case, speed is s + 2, so time walking is 9/(s + 2) hours.
    ↪ Total time is 9/(s + 2) + t/60 = 2.4 hours.

So, we have two equations:

1) 9/s + t/60 = 4

2) 9/(s + 2) + t/60 = 2.4

We can subtract the second equation from the first to eliminate t/60.

So, (9/s + t/60) - (9/(s + </think>

Now, let's solve these equations step by step.

From equation 1:
```

```
9/s + t/60 = 4 --> t/60 = 4 - 9/s --> t = 60*(4 - 9/s) = 240 - 540/s.

From equation 2:
9/(s + 2) + t/60 = 2.4 --> t/60 = 2.4 - 9/(s + 2) --> t = 60*(2.4 - 9/(s
    ↪ + 2)) = 144 - 540/(s + 2).

Now, set the two expressions for t equal:
240 - 540/s = 144 - 540/(s + 2).

Simplify:
240 - 144 = 540/s - 540/(s + 2)
96 = 540*(1/s - 1/(s + 2)).

Simplify the right side:
1/s - 1/(s + 2) = (s + 2 - s)/(s(s + 2)) = 2/(s(s + 2)).

Thus:
96 = 540*(2)/(s(s + 2)) --> 96 = 1080/(s(s + 2)).

Multiply both sides by s(s + 2):
96*s(s + 2) = 1080.

Divide both sides by 24:
4*s(s + 2) = 45 --> $4s^2$ + 8s - 45 = 0.

Solve the quadratic equation:
s = [-8 $\pm$ sqrt(64 + 720)]/(2*4) = [-8 $\pm$ sqrt(784)]/8 = [-8 $\pm$
    ↪ 28]/8.

Since speed can't be negative, s = (20)/8 = 2.5 km/h.

Now, find t from equation 1:
t = 240 - 540/2.5 = 240 - 216 = 24 minutes.

Now, for the third scenario, speed is s + 0.5 = 3 km/h.

Time walking: 9/3 = 3 hours.

Total time including coffee: 3 + t/60 = 3 + 24/60 = 3 + 0.4 = 3.4 hours =
    ↪  3 hours 24 minutes.

Thus, the total time is 3 hours 24 minutes.

The answer is \boxed{204} minutes.
```

Listing 2: Response from E1 model with 1091 tokens.

## H    CHOICE OF TRAINING BUDGETS

We fix the solution budget to 1K tokens because, in practice, we observe that when the `</think>` token is manually inserted, the model naturally transitions to concise solutions that typically require fewer than 1K tokens, even for complex problems. For the thinking budget, we consider four settings $t^* \in \{0.5K, 1K, 2K, 3K\}$. Among these, $t^* = 1K$ achieves the lowest training cost while yielding validation loss comparable to that of larger budgets (2K and 3K).

The ablation study in Section 4.4.2 further supports this choice: the $1K + 1K$ configuration consistently provides the best trade-off between performance and training efficiency. Based on this one-time selection, we adopt $t^* = 1K$ and $s^* = 1K$ as the default setting across all math and code experiments.

## I  MODEL VARIANTS

In this section, we present the results of our method on Skywork-OR1-Math-7B [3]. We fine-tuned it using our budget-constrained rollout method for 200 RL steps with the same hyperparameters as E1-Math-1.5B. The results on AIME24 (Avg@16) under multiple budget settings are shown in Table I.1.

Unconstrained performance increases by +1.3% while requiring fewer tokens. Under strict token budgets, the improvements are both substantial and consistent. The trend mirrors the results observed on E1-Math-1.5B, demonstrating that our method generalizes well across model architectures and training pipelines. E1-Skywork consistently outperforms L1-Skywork across all budget settings.

| Model | Full | 1.5K | 2K | 3K | 4K |
|---|---|---|---|---|---|
| Skywork-OR1-Math-7B | 68.3 (13803) | 1.0 (1534) | 2.1 (2047) | 7.7 (3051) | 14.0 (4023) |
| E1-Skywork | 69.6 (11768) | 16.9 (1381) | 21.3 (1841) | 26.0 (2799) | 32.9 (3742) |
| L1-Skywork-Exact | 59.6 (8277) | 8.1 (1481) | 14.8 (1948) | 24.4 (2805) | 31.7 (3497) |

Table I.1: Performance of Skywork-OR1-Math-7B and E1-Skywork-OR1-Math-7B on AIME under different inference budgets, averaged over 16 runs.

## J  VARYING THE SOLUTION-TOKEN BUDGET

| Model | 1K + 0.25K | 1K + 0.5K | 1K + 0.75K | 1K + 1K | 1.75K + 0.25K |
|---|---|---|---|---|---|
| **E1-Math-1.5B** | 4.8% | 12.9% | 15.8% | **17.5%** | 5.0% |

Table J.1: Effect of varying the solution budget under a fixed 1K thinking budget.

Reducing the solution budget causes a substantial performance drop, even when the thinking phase is preserved. Shrinking the solution phase to 0.25K tokens severely limits performance. Even when increasing the thinking budget from 1K to 1.75K, the gain is negligible (4.8% → 5.0%). This indicates that a minimum solution budget is necessary to reliably articulate the final answer, regardless of how much reasoning is performed. This supports the core motivation of Elastic Reasoning: Reliable performance under strict budgets requires explicitly reserving solution tokens, not just maximizing reasoning length.

## K  ZERO-THINKING (0K + 1K)

| Budget | AIME24 | MATH500 |
|---|---|---|
| **1K + 1K** | **17.5%** | **79.5%** |
| **0K + 1K** | 0.4% | 55.9% |

Table K.1: Performance under zero-thinking (0K + 1K) compared to the standard (1K + 1K) budget.

On **hard problems (AIME24)**, the model is essentially unable to solve tasks without a reasoning phase (17.5% → 0.4%). On **easier tasks (MATH500)**, the model retains some accuracy (55.9%), consistent with prior observations that certain math items can be solved via pattern recognition or shallow heuristics. Overall, this shows that the **thinking phase is crucial for deep reasoning**, while a sufficient solution budget is necessary to reliably express the final answer.

---

[3] https://huggingface.co/Skywork/Skywork-OR1-Math-7B

## L    REPORTING MEAN AND STANDARD DEVIATION

We report results as **mean ± standard deviation** on two datasets: AIME2024 (math reasoning) and LiveCodeBench (coding). For AIME2024, we compare against both L1-Exact and L1-Max. For LiveCodeBench, we report only E1-Code, as the publicly released L1 implementation does not support code fine-tuning.

| Model | 2K + 1K | 3K + 1K |
|---|---|---|
| **E1-Math-1.5B** | $24.8\% \pm 3.7\%$ | $27.9\% \pm 3.3\%$ |
| **L1-Exact** | $14.7\% \pm 3.0\%$ | $20.4\% \pm 3.7\%$ |
| **L1-Max** | $24.0\% \pm 4.6\%$ | $25.8\% \pm 5.7\%$ |

Table L.1: AIME performance under different thinking+solution token budgets.

| Model | 2K + 1K | 3K + 1K |
|---|---|---|
| **E1-Code-1.5B** | $41.6\% \pm 0.98\%$ | $44.1\% \pm 1.3\%$ |

Table L.2: LiveCodeBench performance under different thinking+solution token budgets.

## M    BUDGET-ALIGNED COMPARISON

We provide a budget-aligned comparison in Table M.1. These results evaluate all models under the same inference-time budget. Under matched budgets, E1-Math consistently outperforms L1-Exact, and surpasses L1-Max at medium and larger budgets (3K and 4K). Notably, E1 retains most of the original model's 24K-token performance while requiring far fewer RL training steps (200 vs. 820).

| Inference Budget | 1.5K | 2K | 3K | 4K | 24K |
|---|---|---|---|---|---|
| **L1-Exact** | 4.3% | 10.2% | 14.7% | 20.4% | 24.2% |
| **L1-Max** | 17.5% | 21.3% | 24.0% | 25.8% | 27.1% |
| **E1-Math** | 13.5% | 17.5% | **24.8%** | **27.9%** | **35.0%** |

Table M.1: Pass@1 performance on AIME2024 under matched inference budgets.

At small budgets (up to 2K), L1-Max is slightly stronger. At medium and large budgets (3K and 4K), E1-Math surpasses L1-Max, consistent with its superior test-time scaling behavior. In the unconstrained setting, E1-Math preserves most of the original model's full-budget accuracy (35.0%), while L1-Max shows degradation from overfitting to short reasoning (27.1%). Overall, these results indicate that E1 offers a more stable and scalable approach as reasoning depth increases.

## N    SCALING THINKING VS. SCALING SOLUTION

Below we report both thinking scaling andsolution scaling results under equal total token budgets. All numbers correspond to Pass@1 on AIME24 (Avg@16).

| Model | 1K + 1K | 0K + 2K | 2K + 1K | 0K + 3K |
|---|---|---|---|---|
| **E1-Math-1.5B** | **17.5%** | 15.0% | **24.8%** | 22.5% |
| DeepScaleR-1.5B | 9.6% | 2.9% | 15.8% | 11.4% |

Table N.1: Separate-budget inference under equal total budgets.

**Separate budgeting (our inference mechanism).** Across these equal-budget comparisons, several clear patterns emerge. First, separate budgeting consistently outperforms vanilla truncation,

demonstrating that enforcing a dedicated reasoning phase is crucial for solving challenging problems. Second, explicit reasoning remains beneficial even when the total budget is fixed. For example, at 2K total tokens, the 1K+1K configuration (17.5%) surpasses the 0K+2K configuration (15.0%), and a similar advantage holds at 3K total tokens, where 2K+1K (24.8%) outperforms 0K+3K (22.5%). This indicates that allocating part of the budget to structured thinking is consistently superior to spending all tokens on the final solution.

Third, removing the thinking phase entirely leads to substantial accuracy degradation, especially before training, showing that a clear reasoning segment is essential for deeper mathematical tasks. Fourth, both the thinking and solution budgets influence performance, but increases in the thinking budget lead to noticeably larger gains, suggesting that deeper reasoning is the primary driver of improvement. Finally, the budget-constrained rollout used during training strengthens the scaling behavior of both phases: after training, E1 models exhibit robust improvements in both thinking and solution effectiveness under matched budgets.

## O    DISCUSSIONS

### O.1    WHY IS E1-MATH-1.5B SLIGHTLY BELOW ITS ORIGINAL BASELINE?

We believe the small Math model (1.5B) is more sensitive to perturbations during RL fine-tuning:

**Limited capacity.** Smaller models have less representational capacity and are more prone to catastrophic forgetting when optimized with RL signals. Achieving both strong reasoning and strict budget robustness is more challenging in this size regime.

**Trade-off between robustness and maximal performance.** Our method explicitly trains the model to produce correct answers even when reasoning is prematurely truncated. This robustness objective can slightly reduce maximum unconstrained performance on smaller models, although it improves practical performance under constraints.

### O.2    WHY WE CHOOSE 1K SOLUTION BUDGET?

Before designing the training and evaluation setup, we examined the token distribution of long-reasoning models. As shown in Figure 7, the solution segment averages approximately 445 tokens on AIME24 (math) and 733 tokens on LiveCodeBench (code). Thus, the natural solution length for both tasks is well below 1K tokens.

For this reason, we fixed the solution budget to 1K tokens: it comfortably accommodates the typical solution length across math and code datasets, and it provides a stable experimental configuration that isolates the effect of varying the thinking budget. In contrast, the thinking budget naturally grows much larger and is the primary factor driving test-time scaling behavior, which is why we vary it extensively (1.5K → 4K → 24K).

