# OpenReview forum: "Scalable Chain of Thoughts via Elastic Reasoning"
_ICLR.cc/2026/Conference — ICLR 2026 Poster_

### Official Review · Reviewer_9D8U · 2025-10-20

**Soundness:** 3
**Presentation:** 3
**Contribution:** 3
**Rating:** 6
**Confidence:** 3

**Summary:**

The paper proposes Elastic Reasoning (ER), a framework that makes large reasoning models more efficient and controllable under limited inference budgets. By separating reasoning into thinking and solution phases with independent token budgets and training via budget-constrained reinforcement learning, ER achieves concise yet reliable reasoning—reducing token use by over 30% while maintaining or improving accuracy on math and coding benchmarks.

**Strengths:**

- Novel framework for budget-aware reasoning – The proposed Elastic Reasoning introduces a clear separation between thinking and solution phases, enabling fine-grained control over inference cost without sacrificing performance.
- Strong empirical efficiency and robustness – The method achieves reduction in token usage while maintaining or even improving accuracy on diverse math and coding benchmarks.
- Excellent generalization under unseen budgets – Models trained with a single budget configuration generalize effectively to new inference constraints, demonstrating strong adaptability and practical scalability.

**Weaknesses:**

- The method is only tested on strong reasoning models (DeepScaleR, DeepCoder); it’s unclear whether it generalizes to weaker models like Qwen2.5-Math, which lack explicit CoT structure or strong reasoning priors.
- The paper shows that most improvement comes from the solution phase, while increasing the thinking budget (e.g., 2K–3K tokens) brings little additional gain. This suggests that the model may not truly improve its reasoning efficiency—instead, it might rely on memorized solutions rather than performing deeper thinking.
- The evaluation lacks out-of-domain (OOD) reasoning benchmarks such as MMLU or GPQA. As all experiments focus on math and code, it remains unclear whether Elastic Reasoning generalizes to broader reasoning domains or tasks requiring factual and conceptual knowledge.

**Questions:**

- Can the authors evaluate Elastic Reasoning on out-of-domain reasoning benchmarks (e.g., MMLU, GPQA) to verify whether the method generalizes beyond math and code tasks?
- Can the authors test this by removing the thinking phase entirely (i.e., prompting the model to output only the solution) and reporting the resulting accuracy?
- Could the authors explore asymmetric budgets (e.g., 1.75K for thinking and 0.25K for solution) to test whether the model still maintains performance with a shorter solution phase? Most solutions require far fewer tokens than reasoning.

---

> ### Author Response · Authors · 2025-11-20
> **Response to Reviewer 9D8U (1/n)**
>
> We thank the reviewer for the comments and questions.
>
> >**W1.** It’s unclear whether it generalizes to weaker models like Qwen2.5-Math.
>
> **A.** Thank you for raising this point. Our focus is indeed on *Long-CoT reasoning models*, because these are the models for which **efficiency, controllability, and test-time scaling** become pressing challenges. Weaker models such as Qwen2.5-Math typically:
> - produce **very short reasoning traces**,
> - do **not** exhibit long-horizon CoT explosion, and
> - do **not benefit meaningfully** from split-budget control, as their reasoning is already compact.
>
> For these models, inference-time token budgets are rarely the bottleneck; in contrast, Long-CoT models routinely produce **thousands** of tokens, making length control a critical practical problem that prior work has not addressed effectively.
>
> ### Additional experiment to demonstrate generality
>
> To show that Elastic Reasoning is **not tied to DeepScaleR or DeepCoder**, we added an experiment on another ** Long-CoT model**, *Skywork-OR1-Math-7B* [1]. We fine-tuned it using our method for 200 RL steps with the **same hyperparameters** as E1-Math-1.5B. The results on AIME24 (Avg@16) are shown below.
>
> - **Unconstrained performance improves** (+1.3%) while using **fewer tokens**.
> - **Under strict budget constraints**, the improvements are large and consistent.
> - The trends match those observed for E1-Math-1.5B, confirming that the method generalizes across architectures and training pipelines.
>
> | Model | Tokens | Acc (%) | Tokens | Acc (%) | Tokens | Acc (%) | Tokens | Acc (%) | Tokens | Acc (%) |
> |---------------|--------------|---------------|--------------|---------------|--------------|---------------|--------------|---------------|--------------|---------------|
> | Skywork-OR1-Math-7B | 13803 | 68.3 | 1534 | 1.0  | 2047 | 2.1 | 3051 | 7.7 | 4023 | 14.0 |
> | E1-Skywork-OR1-Math-7B  | 11768 | 69.6 | 1381 | 16.9 | 1841 | 21.3 | 2799  | 26.0 | 3742 | 32.9 |
>
> This additional experiment demonstrates that Elastic Reasoning generalizes across **multiple Long-CoT model families**, not just the two used in the main submission. We will add this discussion and the accompanying results in the revision.
>
> [1] https://huggingface.co/Skywork/Skywork-OR1-Math-7B
>
>
> >**W2.** Increasing the thinking budget (e.g., 2K–3K tokens) brings little additional gain. The model might rely on memorized solutions rather than performing deeper thinking.
>
> **A.** Thank you for raising this important point. We respectfully disagree with your interpretation that (1) “increasing the thinking budget brings little additional gain,” and (2) the model “might rely on memorized solutions.” Our empirical results show the opposite.
>
> **1. Increasing the thinking budget **does** yield substantial gains**
>
> In our experiments, the solution budget is fixed to 1K, so improvements from extending the **thinking** phase can be isolated. On AIME2024, E1-Math-1.5B shows strong positive scaling:
>
> | Model | 0.5K + 1K | 1K + 1K | 2K + 1K | 3K + 1K |
> |--------|------------|------------|-------------|-------------|
> | **E1-Math-1.5B** | 13.5% | 17.5% | 24.8% | 27.9% |
>
> This represents a **+14.4% absolute gain** simply by increasing the reasoning budget. This pattern holds across other math benchmarks (Fig. 4), consistent with known **test-time scaling laws** for reasoning models.
>
> **2. Coding tasks show even stronger thinking-scaling behavior**
>
> As shown in Table 1 of the original paper, extending the thinking budget for E1-Code-14B leads to substantial improvements:
>
> - **LiveCodeBench accuracy:**
>   37.3% → 47.0% when increasing reasoning from 1K → 4K
> - **Codeforces percentile:**
>   78.1 → 92.3
>
> **3. Evidence against the “memorization” hypothesis**
>
> In **Q2**, we provide the results "no thinking": E1-Math-1.5B with 0K + 1K (no thinking) budget only obtains 0.4% on AIME24, confirming answers are not retrieved from memory.
>
> ---
>
> Elastic Reasoning **does** improve reasoning efficiency. The thinking-phase scaling clearly demonstrates deeper reasoning capability rather than memorization. We will clarify this analysis in the revised version.
>
> >**W3.** The evaluation lacks out-of-domain (OOD) reasoning benchmarks such as MMLU or GPQA.
>
> **A.** Thank you for this suggestion. We would like to clarify that our submission **already includes OOD evaluations**. Specifically, Appendix F (Table F.1) reports results on **MMLU**, a broad general-reasoning benchmark that covers factual, conceptual, and multi-disciplinary tasks.
>
> The results show that Elastic Reasoning **generalizes robustly beyond math and code**:
> - Performance improves consistently as the token budget increases,
> - The model maintains stable accuracy even under strict budget constraints, and
> - The positive scaling trend mirrors what we observe on in-domain tasks.
>
> We will make this connection more explicit in the main text to avoid confusion and highlight that OOD evaluation is already included.

---

> > ### Author Response · Authors · 2025-11-20
> > **Response to Reviewer 9D8U (2/n)**
> >
> > >**Q1.** Can the authors evaluate Elastic Reasoning on out-of-domain reasoning benchmarks?
> >
> > **A.** Please refer to **W3**.
> >
> > >**Q2.** Can the authors test this by removing the thinking phase entirely?
> >
> > **A.** Thank you for this helpful suggestion. We ran an additional ablation experiment where we **remove the thinking phase entirely** (`0K + 1K` budget), forcing the model to output only the solution. Below are the Pass@1 results (averaged over 16 runs) on AIME24 and MATH500:
> >
> > | Budget | AIME24 | MATH500 |
> > |--------|--------|----------|
> > | **1K + 1K** | **17.5%** | **79.5%** |
> > | **0K + 1K** | **0.4%** | **55.9%** |
> >
> > - On **hard problems** (AIME24), the model is **unable to solve** the tasks without a reasoning phase—accuracy drops from 17.5% to 0.4%.
> > - On **easier problems** (MATH500), the model still answers some questions correctly without explicit thinking (55.9%), consistent with prior observations that some math queries can be solved via shallow reasoning.
> > - The large drop in both datasets confirms that the **thinking phase is essential** for deep reasoning tasks, and Elastic Reasoning’s ability to preserve structured, compact reasoning under budget constraints is a key contributor to its performance.
> >
> > We will include this ablation in the revised appendix to further support the qualitative findings.
> >
> > >**Q3.** Could the authors explore asymmetric budgets (e.g., 1.75K for thinking and 0.25K for solution).
> >
> > **A.** Thank you for the insightful question. We conducted additional experiments on AIME24 to evaluate *asymmetric* budgets that heavily shrink the solution segment (e.g., 0.25K tokens). The results (Pass@1, averaged over 16 runs) are shown below:
> >
> > | Model | 1K + 0.25K | 1.75K + 0.25K | 1K + 1K |
> > |--------|------------|----------------|----------|
> > | **E1-Math-1.5B** | 4.8% | 5.0% | 17.5% |
> >
> > 1. **Shrinking the solution phase to 0.25K tokens severely limits performance.**
> >    Even when increasing the thinking budget from 1K → 1.75K, the gain is negligible (4.8% → 5.0%).
> >    This indicates that a **minimum solution budget** is necessary to reliably articulate the final answer, regardless of how much reasoning is performed.
> >
> > 2. With the same total budget (2K), 1K budget for solution is much better
> >    Comparing:
> >    - **1K + 1K** → 17.5%
> >    - **1.75K + 0.25K** → 5.0%
> >
> > 3. This supports the core motivation of Elastic Reasoning: Reliable performance under strict budgets requires **explicitly reserving solution tokens**, not just maximizing reasoning length.
> >
> > Most problems require only a small number of tokens to present the final answer, but if the solution segment becomes *too short*, the model cannot finalize its reasoning or output a valid answer—even if it had more thinking time. This validates our design choice of separating budgets and ensuring a dedicated solution budget.
> >
> > We will add these asymmetric-budget results in the revised version.

---

> > > ### Comment · Reviewer_9D8U · 2025-11-26
> > >
> > > Regarding your response to Q2, I have two minor follow-up questions:
> > > 1. Fair comparison under equal total budget.
> > > In the current ablation, you compare 1K + 1K (thinking + solution) with 0K + 1K (no thinking + solution). This changes both the presence of the thinking phase and the total token budget. Would it be possible to also report a comparison between 1K + 1K and 0K + 2K, so that the overall budget is kept constant? This would more cleanly isolate the effect of having an explicit thinking phase versus allocating all tokens to the solution phase.
> > > 2. Choice of a 1K solution budget.
> > > Could you clarify why the solution budget was fixed to 1K tokens? Is this choice supported by empirical observations (e.g., most solutions fit within 1K tokens), prior work, or practical constraints? A brief justification of this design choice would help readers better understand the experimental setup.

---

> > > ### Comment · Reviewer_9D8U · 2025-11-26
> > >
> > > Thanks for the reply of the authors. At this point, I would like to maintain my socre.

---

> > ### Comment · Reviewer_9D8U · 2025-11-26
> >
> > Regarding the authors’ response to W2, I have several follow-up questions and comments.
> >
> > First, in Figure 6, most data points for E1-Math-1.5B (1K+1K) appear to lie near the upper region, suggesting strong performance. However, in the last row of Table 2, the results clearly indicate that increasing the thinking budget leads to meaningful improvements. This seems somewhat contradictory to the visual trend in Figure 6. Could you clarify whether the experimental settings or evaluation metrics differ between Figure 6 and Table 2, which might explain this discrepancy?
> >
> > Second, in Figure 4, L1-Max consistently outperforms E1-Math-1.5B (Budget Constraint). Is this primarily due to the larger model size of L1-Max, which naturally offers stronger reasoning ability? Or are there other contributing factors (e.g., training data scale, optimization strategy, architectural differences) that explain this gap?
> >
> > Finally, I have a little suggestion regarding the figures:
> > instead of using the actual token count on the x-axis, it may be clearer to use the designated thinking budget (e.g., 0.5K, 1K, 2K, 3K).
> > Using raw token counts mixes the thinking phase with the answer phase and makes the visual comparison across different settings more difficult. Using a unified think budget axis would group comparable configurations together, making cross-model comparison and interpretation significantly clearer.

---

> ### Author Response · Authors · 2025-11-28
>
> We sincerely thank the reviewer for acknowledging our efforts during the rebuttal. We are very willing to address the remaining concerns.
>
> >**Q1** Clarify whether the experimental settings or evaluation metrics differ between Figure 6 and Table 2?
>
> **A.** Thank you for highlighting this — the difference comes from the fact that **Figure 6 and Table 2 evaluate two different dimensions of the method**.
>
> ### 1. What Figure 6 measures
> Figure 6 reports an **ablation on the *training* budgets**, i.e., how many thinking/solution tokens are allowed **during RL rollout training**.
> These results show that: Using **1K thinking + 1K solution during training** provides the most stable and effective learning signal. Thus, Figure 6 answers the question:
> *“What training budget should we use for the RL rollout?”*
>
> ### 2. What Table 2 measures
> The last row of Table 2 evaluates **test-time (inference) scaling** of the **trained E1 model** by varying only the *inference* thinking budget (1K → 3K), while keeping training fixed.
>
> These results demonstrate: A clear positive **test-time scaling law**: more thinking tokens at inference → higher accuracy.
>
> Thus, Table 2 answers the question: *“After training is fixed, what happens when we increase the inference thinking budget?”*
>
> - **Figure 6**: compares different *training* setups, not inference scaling.
> - **Table 2**: keeps training fixed (at 1K+1K) and varies *inference* budgets.
>
> These two observations are consistent and address different aspects of the method (training vs. inference).
>
>
> >**Q2.** Explain why L1-Max consistently outperforms E1-Math-1.5B?
>
> **A2.** Thank you for the question, we are happy to clarify this. The performance gap in Figure 4 is not due to model size, architecture. All models compared in Figure 4 use the same base model (DeepScaleR-1.5B). The key differences come from the **training strategy and degree of RL fine-tuning**.
>
> ### 1. L1-Max undergoes much heavier RL training
> L1-Max is trained with **820 RL steps**, specifically optimized to follow a fixed thinking-length instruction.
> This extensive training:
> - sharpens its behavior under **small thinking budgets**,
> - but causes the model to **overfit** to short reasoning patterns,
> - and leads to a **significant drop in performance in unconstrained settings**.
>  - **L1-Max:** 27.1%
>  - **E1-Math-1.5B:** 35.0%
>
> Thus, L1-Max performs better than E1 **only in very small budgets**, but at the cost of *substantially degrading* the model’s overall reasoning ability.
>
> ### 2. E1 preserves the model’s original reasoning ability ans scales much better.
> E1-Math-1.5B uses only **200 RL steps**, with a training objective matched to the 1K+1K rollout.
> This conservative and aligned training:
> - maintains the original 35.0% unconstrained accuracy,
> - offers strong performance under small and medium budgets,
> - and **surpasses L1-Max** when the thinking budget increases.
>
> In the results under the same inference budgets (in Q3): E1-Math surpasses L1-Max at medium and larger budgets (3K and 4K). This shows that L1-Max is optimized for **short thinking lengths**, while E1 scales much better as thinking tokens increase.
>
>
> >**Q3.** Suggestions for using thinking budget for x-axis.
>
> **A.** Thank you for the helpful suggestion regarding the x-axis design. We agree that using the *designated thinking budget* would make cross-model comparisons clearer, as it groups comparable configurations more intuitively.
>
> To complement your suggestion, we also provide a **budget-aligned comparison** below. These results directly compare models under the *same inference budget*. Under matched budgets, **E1-Math consistently outperforms L1-Exact**, and surpasses **L1-Max** at medium and larger budgets (3K and 4K). Importantly, E1 retains most of the original model’s 24K-token performance while requiring far fewer RL training steps (200 vs. 820).
>
> | Inference Budget | 1.5K | 2K | 3K | 4K | 24K |
> |------------------|---------|--------|--------|--------|--------|
> | **L1-Exact** | 4.3% | 10.2% | 14.7% | 20.4% | 24.2% |
> | **L1-Max**   | 17.5% | 21.3% | 24.0% | 25.8% | 27.1% |
> | **E1-Math**  | 13.5% | 17.5% | **24.8%** | **27.9%** | **35.0%** |
>
> - **Small budgets (≤2K):** L1-Max is slightly higher due to heavy (820-step) RL tuning to short reasoning traces.
> - **Medium/Large budgets (≥3K):** E1-Math surpasses L1-Max, consistent with the test-time scaling trends.
> - **Unconstrained:** E1-Math preserves most of the original model’s accuracy (35.0%), while L1-Max suffers from over-fitting to short reasoning (27.1%).
>
> This reinforces that E1 offers a more stable and scalable approach, particularly when reasoning depth increases.
>
> We appreciate the reviewer’s suggestion and will include in the revision to use the cleaner budget axis for improved interpretability.

---

> ### Author Response · Authors · 2025-11-28
>
> >**Q4.** Fair comparison under equal total budget.
>
> **A.** Thank you for the insightful suggestion. We agree that comparing **1K+1K** with **0K+2K** (equal total budget = 2K) provides a cleaner way to isolate the effect of having an explicit thinking phase.
>
> Below we report both **vanilla truncation** and **separate budgeting** results under equal total budgets. All numbers are Pass@1 on AIME24 (Avg@16).
>
> ---
>
> ### **1. Vanilla Truncation (no separate thinking/solution phases)**
> Here the model generates freely until the token limit is reached.
>
> | Model | **2K** | **3K** |
> |--------|------------------|------------------|
> | DeepScaleR-1.5B | 1.4% | 5.2% |
>
> Vanilla truncation performs extremely poorly because the model often consumes most of the budget in incomplete or irrelevant reasoning before reaching the final answer.
>
> ---
>
> ### **2. Separate Budgeting (our inference mechanism)**
>
> | Model | **1K + 1K** | **0K + 2K** | **2K + 1K** | **0K + 3K** |
> |--------|--------------|--------------|--------------|--------------|
> | **E1-Math-1.5B** | **17.5%** | 15.0% | **24.8%** | 22.5% |
> | DeepScaleR-1.5B | 9.6% | 2.9% | 15.8% | 11.4% |
>
> ### **Key Findings**
> Across our equal-budget comparisons, several consistent findings emerge:
>
> 1. **Separate budgeting far outperforms vanilla truncation.**
>
> 2. **Explicit reasoning is beneficial even with fixed total budget.**
>    At 2K total tokens, **1K+1K (17.5%) > 0K+2K (15.0%)**;
>    at 3K total tokens, **2K+1K (24.8%) > 0K+3K (22.5%)**.
>    This shows that allocating part of the budget to a dedicated thinking phase is consistently superior to spending all tokens on the solution.
>
> 3. **Without training, zero thinking performs substantially worse.**
>    Removing the thinking phase before training leads to **severe accuracy degradation**, highlighting that a clear reasoning segment is essential for deep problems.
>
> 4. **Both thinking and solution budgets matter, but thinking scales better.**
>    Increasing the solution budget improves performance, but increasing the thinking budget yields **larger gains**, indicating that deeper reasoning is the primary driver of improvement.
>
> 5. **Our budget-constrained rollout boosts both thinking and solution scaling.**
>    After training, E1 models show strong scaling in both dimensions.
>
> We will include these insights and the equal-budget comparisons in the revised version, as they more clearly isolate and demonstrate the effectiveness of separate budgeting.
>
> >**Q5.** Clarify why the solution budget was fixed to 1K tokens.
>
> **A.** Thank you for the question. We are happy to clarify our choice of a 1K solution budget.
>
> Before designing the training and evaluation setup, we analyzed the **token distribution** of long-reasoning models. As shown in Figure 7 of the main paper:
>
> - The **solution segment** averages **445 tokens** on AIME24 (math).
> - The **solution segment** averages **733 tokens** on LiveCodeBench (code).
>
> Thus, the natural solution length for these tasks is well under 1K tokens.
>
> Based on this empirical observation, we chose **1K** as the solution budget because:
>
> 1. **It comfortably covers the typical solution length** for both math and code tasks.
> 2. **It keeps the setup consistent** across all experiments, allowing us to focus on the effect of varying the *thinking* budget.
>
> In contrast, the **thinking** budget naturally grows much larger and is the main driver of test-time scaling, which is why we vary it extensively (1.5.5K → 4K → 24K).
>
> We will make this design rationale explicit in the revision.

---

### Official Review · Reviewer_apFU · 2025-10-27

**Soundness:** 2
**Presentation:** 2
**Contribution:** 2
**Rating:** 2
**Confidence:** 3

**Summary:**

The paper proposes Elastic Reasoning, a method that enables large reasoning models to achieve scalable and adaptive length control. To further improve solution quality under incomplete reasoning, we introduce a novel training strategy called budget-constrained rollout, which teaches the model to generate high quality answers even with partial CoT trajectories. This method is integrated into GRPO training. This method produces E1-Math-1.5B and E1-Code-14B.

**Strengths:**

- Very simple method that solves the problem of truncated solution in long reasoning.
- The thinking-solution ablation (4.4.1) is interesting and is good evidence to understand what the proposed training method improves (i.e., generating a solution under an incomplete thinking process).

**Weaknesses:**

## Major

- Figure 1, 4, 5, 6 are a bit unclear (This may be a minor weakness, but I assigned this as a major weakness for now because it is a crucial experimental setup):
    - What are the points? Do they correspond to the whole AIME questions across different budgets?
    - What is the x-axis? Is it the average tokens used?
    - Could you include the error bars (x- and y-axis)? This is particularly important as there are cases where the Pass@1 and tokens used are not significantly different
- Lack of model variations
    - The authors only experimented with one model variant per setup (one for math, and one for coding).
    - I understand that such experiments can be costly. However, I cannot confidently argue for the generalizability of this finding.
- Lack of further analyses
    - I am curious about the qualitative difference between the thinking process before and after the training (e.g., do the models after fine-tuning commit less backtracking? less circular reasoning?)

## Minor

- Missing details of the GRPO training
    - What is the reward design? It is unclear whether when the training “converges” around 0.5 reward score is a good thing or not.
    - What about the other hyperparameters?
    - Please generally be thorough in describing the experiment

## Additional Suggestions

- Typo Section 4.2 title

**Questions:**

- Just to confirm that I am understanding the novelty correctly: Two phases of thinking and solution seems to be exactly what reasoning models like DeepSeek is doing, right? Am I missing a certain novelty claim by the authors here? Is it simply that prior works were enforcing the overall tokens count, but now budget-constrained rollout limits the thinking tokens count separately from the solution token?
- Are there qualitative difference between the reasoning patterns of naive truncation vs Elastic Reasoning?
- Appendix B mentioned that the training is conducted for only 30 steps, but that does not seem to match Figure 3.
- Any intuition why the performances of the trained models are still lower than the original model’s?
- Have you tried varying the solution budget instead? or perhaps a control experiment with 0 thinking token and 1K solution tokens may be interesting.

---

> ### Author Response · Authors · 2025-11-20
> **Response to Reviewer apFU (1/n)**
>
> We thank the reviewer for the comments and questions.
>
> >**W1.** Clarify Figure 1, 4, 5, 6.
>
> **A.** Thank you for pointing out the clarity issues in Figures 1, 4, 5, and 6. We will revise these figures to make the setup more explicit.
>
> - **What do the points represent?**
>
> Each point corresponds to the *Pass@1 performance of a model on the entire AIME2024 set* evaluated under a specific inference-time token budget.
>
> - **What is the x-axis?**
>
> The x-axis reports the **average number of tokens used per problem**, measured under the corresponding budget configuration. This matches the evaluation protocol used by our main baseline L1.
>
> - **Why no error bars?**
>
> We followed the experimental setup of L1, which reports averaged Pass@1 without variance bars. All results shown are *averages over 16 runs* for math tasks (and 8 for code tasks).
>   Due to time constraints in the rebuttal period, we may not be able to regenerate every plot with full x- and y-axis error bars, but we fully agree with the reviewer that visualizing variance would strengthen the presentation and we will include these in the furture.
>
> We appreciate the reviewer highlighting this, and we will ensure the revised figures include clearer descriptions.
>
> >**W2.** Lack of model variations
>
> **A.** Thank you for raising this concern. We agree that demonstrating robustness across multiple model backbones is important for establishing generalizability.
>
> To address this, we conducted an additional experiment using another **independently developed Long-CoT model**, *Skywork-OR1-Math-7B* [1]. We fine-tuned this model using our budget-constrained rollout method for 200 RL steps with the **same hyperparameters** as E1-Math-1.5B. The results on AIME24 (averaged over 16 runs) under various budget settings are shown below.
>
> - **Without budget constraints**, the fine-tuned E1-Skywork variant *outperforms the original model by +1.3% accuracy while using fewer tokens.*
> - Under **strict budget constraints**, the improvements are even more pronounced, despite also giving the baseline a fair comparison by using separate budgeting during inference.
> - These results mirror the improvements observed for E1-Math-1.5B, providing strong evidence that our method generalizes across different architectures and training pipelines.
>
>
> | Model | Tokens | Acc (%) | Tokens | Acc (%) | Tokens | Acc (%) | Tokens | Acc (%) | Tokens | Acc (%) |
> |---------------|--------------|---------------|--------------|---------------|--------------|---------------|--------------|---------------|--------------|---------------|
> | Skywork-OR1-Math-7B | 13803 | 68.3 | 1534 | 1.0  | 2047 | 2.1 | 3051 | 7.7 | 4023 | 14.0 |
> | E1-Skywork-OR1-Math-7B  | 11768 | 69.6 | 1381 | 16.9 | 1841 | 21.3 | 2799  | 26.0 | 3742 | 32.9 |
>
> These results further confirm that the proposed *Elastic Reasoning* framework is not specific to a single model but is broadly applicable.
>
> [1] https://huggingface.co/Skywork/Skywork-OR1-Math-7B
>
> >**W3.** Missing details of the GRPO training and explaining the reward curve.
>
> **A.** Thank you for pointing this out. We agree that the GRPO training details should be described more thoroughly. We will expand Section 3.2.3 and Appendix B in the revision. Below we clarify the reward design, training behavior, and hyperparameters.
>
> **Reward design.**
>
> We follow the reward function used in **DeepScaleR-1.5B**, which is also standard in recent Long-CoT RL works:
> - **1** — if the final answer is correct according to our automatic LaTeX/Sympy checker.
> - **0** — otherwise (incorrect answer, malformed output, or missing `<think>` / `</think>` delimiters).
>
> Importantly, since our rollout is **budget-constrained**, the reward is computed based on the trajectory generated under the 1K+1K budget.
>
> **Interpreting the reward curve.**
>
> In binary-reward settings, the *absolute scale* of the average reward is less informative than its *trend*.
> - A reward of ~0.5 during training **does not imply poor performance**; it corresponds to ~50% of sampled rollouts being correct under the strict budget constraint.
> - As shown in Fig. 3 (left), the *Pass@1* metric increases substantially during training, which is the more meaningful measure of reasoning capability.
>
> We will clarify this point explicitly in the revised manuscript.
>
> **Hyperparameter settings.**
>
> We follow **exactly the same hyperparameters** as DeepScaleR-1.5B for fair comparison, except for our modified rollout procedure:
>
> - Batch size: **128**
> - Learning rate: **1e-6**
> - Rollout sampling temperature: **0.6**
> - Number of rollouts per prompt: **8**
> - Budget-constrained rollout: **1K thinking tokens, 1K solution tokens**
> - Total RL steps: **200** (math), **30** (code)
>
> **Reproducibility.**
>
> We provide full training and evaluation code in the supplementary material, enabling replication of all experiments.
>
> We appreciate the reviewer highlighting this and will incorporate a more thorough description of the training setup in the revision.

---

> > ### Author Response · Authors · 2025-11-20
> > **Response to Reviewer apFU (2/n)**
> >
> > >**Q1.** Understanding the novelty correctly.
> >
> > **A.** Thank you for the question. You are correct that many recent reasoning models (e.g., DeepSeek-R1) already *structure* their outputs into `<think>` and `<solution>` segments. Our contribution is not the introduction of two segments, but rather **how these two segments are controlled, trained, and used for inference**. We clarify the novelty below.
> >
> > ### Novelty 1: *Separate* budgets for thinking vs. solution
> > Prior work—including L1—enforces a **single overall length constraint**. In contrast, our work is the first to introduce a **principled formulation of split budgets**:
> > - a thinking budget \(t\), and
> > - a solution budget \(s\),
> >
> > with independent control during inference.
> > This is not done in DeepSeek-R1 or other prior reasoning models, which simply generate two segments but do **not** constrain them separately or treat them as distinct optimization targets.
> >
> > ### Novelty 2: Budget-constrained rollout (training) that matches inference
> > Existing RL approaches (e.g., L1) train on **full trajectories** and then enforce length limits only at inference.
> > Our method instead:
> > - enforces the separate budgets **inside the rollout**,
> > - forces early truncation during training exactly as it will happen at test time, and
> > - teaches the model to produce high-quality solutions even when thinking is incomplete.
> >
> > This budget-matched training is new, and it is key to enabling strong performance under strict token limits.
> >
> > ### Novelty 3: Prompt-agnostic length control
> > Unlike L1 and other prompt-based length-control methods:
> > - we do **not** provide any length instructions or special control tokens,
> > - the model is **never informed** of the budget,
> > - all control happens at the system level via forced `</think>` insertion.
> >
> > This avoids prompt overfitting and enables generalization to unseen budgets.
> >
> > ### Novelty 4: Efficiency and performance preservation
> > Our method achieves:
> > - **200 RL steps** for Math (vs. 700–820 for L1 variants),
> > - significantly **better control–performance tradeoffs**,
> > - and stronger preservation of unconstrained accuracy (e.g., 35.0% vs. 27.1% for L1-Max on AIME2024).
> >
> > This shows the approach is both *more efficient* and *more stable* compared to existing baselines.
> >
> > We will clarify this distinction in the revision.
> >
> > >**Q2.** Qualitative difference between the reasoning patterns of naive truncation vs Elastic Reasoning.
> >
> > **A.** Yes — there are clear qualitative differences between naïve truncation and Elastic Reasoning. We include side-by-side examples in Appendix G (AIME2024, 1.5k token budget), and summarize the distinctions below.
> >
> > ### Naive truncation
> > - The original model often produces **long, verbose reasoning chains**.
> > - When truncated, the chain becomes **incomplete**, often cutting off before reaching the solution.
> > - This typically results in **invalid or missing final answers**, since the solution segment is frequently truncated away.
> >
> > ### Elastic Reasoning (after budget-constrained training)
> > - The model learns to allocate its limited thinking tokens **more purposefully**.
> > - Reasoning becomes **denser, more structured**, and focuses on essential steps rather than verbosity.
> > - The transition into the solution phase is **smoother and earlier**, preserving the answer even under strict budgets.
> > - In the example shown in Appendix G:
> >   - **Original model**: 1537 tokens, verbose reasoning.
> >   - **E1 model**: 1091 tokens, more compact reasoning *and* a correct final answer.
> >
> > Overall, Elastic Reasoning does not merely shorten the chain—it changes the *pattern* of reasoning. The model learns to front-load critical reasoning steps, avoid unnecessary exploration, and reliably produce solutions even under tight token constraints.
> >
> > >**Q3.** Appendix B mentioned that the training is conducted for only 30 steps, but that does not seem to match Figure 3.
> >
> > **A.** Thank you for the question.
> >
> > Appendix B states that:
> > - **E1-Math-1.5B** is trained for **200 RL steps**, and
> > - **E1-Code-14B** is trained for **30 RL steps**.
> >
> > Figure 3 shows the **training curves for E1-Math-1.5B**, which is why the x-axis extends to 200 steps. Thus, there is no mismatch.

---

> ### Author Response · Authors · 2025-11-20
> **Response to Reviewer apFU (3/n)**
>
> >**Q4.** Any intuition why the performances of the trained models are still lower than the original model’s?
>
> **A.** Thank you for the insightful question. We summarize our findings below.
>
> First, only **E1-Math-1.5B** shows a slight drop in *unconstrained* performance compared to its original model. Importantly:
> - Under **budget-constrained inference**, E1-Math-1.5B performs *significantly better* than the original model (Fig. 1).
> - It also performs *better than all L1 variants* in both constrained and unconstrained settings.
>
> In contrast:
> - **E1-Code-14B** actually *improves* over the original DeepCoder-14B in unconstrained settings (Table 1).
> - **E1-Skywork-OR1-Math-7B** (added in the rebuttal) also exceeds the original model in unconstrained performance while using fewer tokens.
>
> ### Why is E1-Math-1.5B slightly below its original baseline?
> We believe the small Math model (1.5B) is more sensitive to perturbations during RL fine-tuning:
>
> 1. **Limited capacity.**
>    Smaller models have less representational capacity and are more prone to catastrophic forgetting when optimized with RL signals. Achieving both *strong reasoning* and *strict budget robustness* is more challenging in this size regime.
>
> 2. **Trade-off between robustness and maximal performance.**
>    Our method explicitly trains the model to produce correct answers *even when reasoning is prematurely truncated*. This robustness objective can slightly reduce maximum unconstrained performance on smaller models, although it improves practical performance under constraints.
>
> Overall, the phenomenon appears size-dependent rather than inherent to the method.
>
> We will clarify this intuition in the revised manuscript.
>
> >**Q5.** Vary the solution budget and experiment with 0 thinking token and 1K solution tokens.
>
> **A.** Thank you for the suggestion. We performed two additional sets of experiments:
>
> 1. **Varying the solution budget while fixing the thinking budget at 1K**
> 2. **Zero-thinking (0K + 1K) control experiments** on both AIME24 and MATH500.
>
> All results are Pass@1 averaged over 16 runs.
>
> ---
>
> ### 1. Varying the solution budget (fixed 1K thinking)
>
> | Model | 1K + 0.25K | 1K + 0.5K | 1K + 0.75K | 1K + 1K |
> |--------|------------|------------|--------------|------------|
> | **E1-Math-1.5B** | 4.8% | 12.9% | 15.8% | **17.5%** |
>
>
> Reducing the solution budget causes a **substantial performance drop**, even when the thinking phase is preserved. This highlights that:
> - The solution phase requires a **minimum token budget** to properly articulate the final answer.
> - Extending the solution budget is consistently beneficial.
>
> This strongly supports our formulation where the solution budget is explicitly preserved and independently controlled.
>
> ---
>
> ### 2. Zero-thinking (0K + 1K) control experiment
>
> | Budget | AIME24 | MATH500 |
> |--------|---------|----------|
> | **1K + 1K** | **17.5%** | **79.5%** |
> | **0K + 1K** | 0.4% | 55.9% |
>
> - On **hard problems (AIME24)**, the model essentially cannot solve tasks without a reasoning phase (17.5% → 0.4%).
> - On **easier tasks (MATH500)**, the model retains some accuracy under zero-thinking (55.9%), consistent with known results that certain math items can be solved via pattern recognition or shallow reasoning.
> - Overall, this confirms that the **thinking phase is essential for deep reasoning**, while a sufficient solution budget is necessary to reliably express the final answer.
>
> ---
>
> - Both experiments illustrate that **thinking and solution phases serve distinct roles**, and **both must be budgeted**.
> - The solution budget is not simply “small but optional”—it materially affects accuracy.
> - Elastic Reasoning is effective because it provides **independent, dedicated budgets** for these two phases and trains the model to use them efficiently.
>
> We will include these results and interpretations in the revised appendix.
>
> >**Suggestion.** Typo Section 4.2 title
>
> **A.**  Thank you for the suggestion. We will fix it in the revision.

---

> > ### Comment · Reviewer_apFU · 2025-11-20
> >
> > Thank you for the thorough response! I decided to raise my score (2→4). Before raising the score further, I would like the authors to address the following points:
> >
> > - Error bars/Statistical rigor
> >     - Apologies for being slightly demanding, but would it be possible for you to run the experiment with perhaps a smaller number of runs and on a couple of datasets (one math and one coding), just enough to understand the variance of the results? If time is still a constraint, perhaps only comparing against the second-best baseline. Please let me know if this is still infeasible. At the moment, it is difficult for me to believe in the results with a limited degree of statistical rigor.
> > - Model variations
> >     - Thank you for providing these results! Apologies for being demanding again, but this claim of improvement makes sense if you also compare it against other methods. Have you tried to compare against L1-trained Skywork?
> >     - Could you also comment on why choosing Skywork? I am not particularly aware of the model’s unique property that would benefit the experimental setup.
> > - Novelty
> >     - Thank you for clarifying the novelty! This is very helpful. Just a follow-up question of lower priority: does that mean E1 is a framework to train the model to work better on S1? My current understanding is that the novelty is a bit limited because of this, but I would not discredit it, as you have provided a good number of empirical results.
> > - Varying solution budget
> >     - Thank you for providing these additional results. It could be interesting to establish the lower bound of both reasoning and solution token budgets. But this is of lower priority for now!
> >
> > It would be amazing if you could highlight the key changes in the revised manuscript so that we, the reviewers, can see the important updates. Thanks again! Looking forward to further discussion!

---

> ### Author Response · Authors · 2025-11-27
> **Response to Reviewer apFU (1/n)**
>
> We sincerely thank the reviewer for acknowledging our efforts during the rebuttal and for raising the score from 2 to 4. We are very willing to address the remaining concerns.
>
> >**Q1.** Error bars/Statistical rigor
>
> **A.** Thank you for the request. We agree that presenting variance is important for interpreting the robustness of our results.
> Below we report **mean ± standard deviation** on two datasets:
> - **AIME2024** (math reasoning)
> - **LiveCodeBench** (coding)
>
> For AIME2024, we compare against both L1-Exact and L1-Max. For LiveCodeBench, we only report E1-Code because the publicly released L1 implementation does not support code fine-tuning.
>
> | Model | 2K + 1K | 3K + 1K |
> |--------|------------|----------------|
> | **E1-Math-1.5B** | 24.8% $\pm$ 3.7% | 27.9% $\pm$ 3.3%|
> | **L1-Exact** | 14.7%$\pm$ 3.0% | 20.4%$\pm$ 3.7% |
> | **L1-Max** | 24.0%$\pm$ 4.6% | 25.8%$\pm$ 5.7% |
>
> | Model | 2K + 1K | 3K + 1K |
> |--------|------------|----------------|
> | **E1-Code-1.5B** | 41.6%$\pm$ 0.98% | 44.1%$\pm$ 1.3% |
>
> >**Q2.** Model variations
>
> **A.** Thank you for the thoughtful follow-up. We have now included **L1-trained Skywork** as an additional baseline to make the comparison fully fair.
>
> To ensure consistency, we used the **original L1 implementation**, changing only the base model to *Skywork-OR1-Math-7B*. The results on AIME24 (Avg@16) under multiple budgets are shown below.
>
> | Model | Tokens | Acc (%) | Tokens | Acc (%) | Tokens | Acc (%) | Tokens | Acc (%) | Tokens | Acc (%) |
> |---------------|--------------|-----------|--------------|-----------|--------------|-----------|--------------|-----------|--------------|-----------|
> | Skywork-OR1-Math-7B | 13803 | 68.3 | 1534 | 1.0 | 2047 | 2.1 | 3051 | 7.7 | 4023 | 14.0 |
> | **E1-Skywork** | 11768 | **69.6** | 1381 | **16.9** | 1841 | **21.3** | 2799 | **26.0** | 3742 | **32.9** |
> | **L1-Skywork-Exact** | 8277 | 59.6 | 1481 | 8.1 | 1948 | 14.8 | 2805 | 24.4 | 3497 | 31.7 |
>
> - **E1-Skywork consistently outperforms L1-Skywork** across all budget settings.
> - Even in *unconstrained* mode, E1-Skywork improves accuracy while also reducing average tokens.
> - Under strict budget constraints, Elastic Reasoning delivers substantially higher accuracy, validating the generality of our approach.
>
> ---
>
> ### Why choose Skywork-OR1-Math-7B?
> We selected Skywork for three reasons:
>
> 1. Fully open-sourced training pipeline.
>    Skywork provides complete access to *model weights, training data, and training scripts*. This makes it an ideal candidate for rigorous and reproducible RL/CoT experiments.
>
> 2. Strong Long-CoT performance.
>    Skywork-OR1-Math-7B achieves **68.3% on AIME24**, placing it among the strongest open-source math models in its size category.
>
> 3. Extremely long reasoning traces.
>    The model naturally produces **very long responses** (avg. 13,803 tokens per question in our evaluation).
>    This property makes Skywork a **stress test** for length-control methods, since uncontrolled Long-CoT generation leads to severe token explosion.
>
> For these reasons, Skywork provides a meaningful and challenging testbed for evaluating the generality of Elastic Reasoning.
>
> We will add these clarifications and the new results to the revised appendix.
>
> >**Q3.** Novelty
>
> **A.** Thank you for the helpful clarification question. We agree that E1 and S1 share a similar *goal*—controlling test-time scaling—but **E1 is not simply a training framework that makes the model better at S1**. The two methods differ in both mechanism and capability.
>
> ### 1. E1 outperforms S1 even *without* any training
> As shown in Figure 1, the **inference-time mechanism alone** (separate budgeting) already produces substantially better performance than S1.
> This indicates that E1 is *not* just “training S1”; the **inference rule itself is stronger and more stable**.
>
> ### 2. Mechanistic difference: S1 controls only thinking; E1 controls *both* thinking and solution
> - **S1** uses **prompt instructions** to limit the *thinking* length, but does not control or reserve tokens for the *solution*.
> - **E1**, by contrast, introduces **two explicitly separated budgets**:
>   - a thinking budget \(t\), and
>   - a solution budget \(s\),
>   enforced via system-level truncation rather than prompts.
>
> This yields **much more precise and reliable control** over total output length, especially under strict token limits.
>
> These differences lead to the large empirical gains observed in the paper.

---

> ### Author Response · Authors · 2025-11-27
> **Response to Reviewer apFU (2/n)**
>
> >**Q4.** Varying solution budget
>
> **A.** Thank you for the suggestion. From our empirical observations:
>
> ### **Training**
> As shown in Section 4.4.2, a budget of **1K thinking + 1K solution** provides a good balance between:
> - allowing sufficient reasoning steps during rollout,
> - leaving room for a complete solution segment.
>
> ### **Inference**
> - The **solution budget** is best kept around **1K**, as our ablations show that reducing it substantially harms accuracy even when the thinking budget is large.
> - The **thinking budget** can be adjusted based on performance requirements and available compute.
>   Our results show strong positive test-time scaling: increasing thinking budget beyond 1K (e.g., 2K → 3K) yields consistent gains on both math and code benchmarks.
>
> >**Q5.** Highlight the key changes
>
> **A.** We have updated the revision and highlighted the changes.

---

### Official Review · Reviewer_8Dnp · 2025-10-31

**Soundness:** 3
**Presentation:** 3
**Contribution:** 3
**Rating:** 6
**Confidence:** 3

**Summary:**

This paper propose a method to control the reasoning sequence length of large language models. The key idea is to have a separate "token budget" for the c-o-t and for the answer, so that when the budget is exhausted for the c-o-t, the model can still produce an answer. Training with GRPO uses rollouts produced with that process, which leads the model to learn to deal with limited budget. The method outperforms alternatives that either limit the complete sequence or train the model to generate tokens that "terminate" its reasoning.

**Strengths:**

Token budget is a key issue for reasoning, the method is very simple and sensical, performance are great.

**Weaknesses:**

Part 3.2.3 could probably be clarified, in particular the authors should provide a clearer description of the quantities involved and in particular the meaning of the conditioning in the policy, and the differences with a vanilla GRPO procedure.

**Questions:**

- How is the model informed of the budget during inference? simply because </think> is generated? Hence the model has no information about the budget before it actually exhausts it? Fig 2 gives the impression that additional tokens specify it (red squares)?

- This is not my direct domain of expertise, so are the baselines considered in the experimental part the best available?

---

> ### Author Response · Authors · 2025-11-20
> **Response to Reviewer 8Dnp**
>
> We thank the reviewer for the comments and questions.
>
> >**W1.** Clarify Part 3.2.3.
>
> **A.** Thank you for the helpful suggestion. We will clarify Section 3.2.3 in the revision.
>
> The key difference between our method and vanilla GRPO lies only in the rollout procedure. In standard GRPO, the policy generates a full trajectory without any structural constraints. In contrast, our budget-constrained rollout conditions the policy on a fixed thinking/solution budget pair (t*, s*). During rollout, the policy generates the thinking segment up to at most t* tokens; if the model does not emit </think> before reaching t*, we forcibly terminate the reasoning by inserting this token. The model then generates the solution segment with a remaining budget of s* tokens. The GRPO objective, advantage normalization, and optimization procedure remain unchanged.
>
> Thus, the conditioning simply indicates that trajectories are sampled under these rollout constraints, rather than from unconstrained generation. This modification allows training to faithfully match the inference-time separate-budgeting mechanism. We will revise Section 3.2.3 to make these quantities and distinctions more explicit.
>
> >**Q1.** How is the model informed of the budget during inference?
>
> **A.** Thank you for the question. We apologize for the misleading presentation in Fig.2 and will clarify this in the revision.
>
> Our method **does not provide the model with any explicit budget information during inference**. Unlike prompt-based length-control approaches (e.g., L1), we do **not** insert additional tokens or instructions indicating how many tokens the model is allowed to generate. Instead, **all length control is enforced externally at the system level** through the *separate-budgeting* mechanism:
>
> - The model begins generating inside a `<think>` block.
> - We enforce a thinking budget \(t\) **outside** the model: if the model does not produce `</think>` before reaching \(t\) thinking tokens, we forcibly insert `</think>` to terminate the reasoning phase.
> - The model then generates the solution segment under a similarly enforced solution budget \(s\).
>
> Thus, the model has **no awareness of the budgets** \(t\) or \(s\) beforehand. It simply continues generation after we insert the `</think>` token.
>
> This design avoids prompt-overfitting issues and ensures that the model generalizes robustly to unseen budget configurations at inference time.
>
> Regarding the red squares in Fig. 2: these are **visual markers** to illustrate different budget settings, and **not** additional tokens supplied to the model. We will revise the caption to prevent this confusion.
>
> >**Q2.** Are the baselines considered in the experimental part the best available?
>
> **A.** Thank you for the question. Our work focuses on *Long-CoT reasoning under strict token budget constraints*, a setting that has only recently begun receiving attention. The availability of strong and reproducible baselines is therefore limited. For context:
> - The earliest open-source Long-CoT models (e.g., DeepSeek-R1) were released only in late **January 2025**.
> - Most follow-up open-source implementations and training pipelines appeared in **February–March 2025**.
> - Research specifically addressing **budgeted or length-constrained Long-CoT reasoning** is even more recent.
> - Among these, **L1** is the most representative and widely adopted RL-based baseline for explicit length control, so we include both L1-Exact and L1-Max.
>
> We evaluated a broad set of *state-of-the-art reasoning models* (e.g., DeepScaleR-1.5B and DeepCoder-14B) to demonstrate that our method generalizes well across different architectures and domains (math and code). We agree that new baselines continue to emerge rapidly, so we added an experiment on *Skywork-OR1-Math-7B* [1]. We fine-tuned it using our budget-constrained rollout method for 200 RL steps with the **same hyperparameters** as E1-Math-1.5B. The results on AIME24 (Avg@16) under multiple budget settings are shown below.
>
> - **Unconstrained performance improves** (+1.3%) while using **fewer tokens**.
> - **Under strict budget constraints**, the improvements are large and consistent.
> - The trends match those observed for E1-Math-1.5B, confirming that the method generalizes across architectures and training pipelines.
>
> | Model | Tokens | Acc (%) | Tokens | Acc (%) | Tokens | Acc (%) | Tokens | Acc (%) | Tokens | Acc (%) |
> |---------------|--------------|---------------|--------------|---------------|--------------|---------------|--------------|---------------|--------------|---------------|
> | Skywork-OR1-Math-7B | 13803 | 68.3 | 1534 | 1.0  | 2047 | 2.1 | 3051 | 7.7 | 4023 | 14.0 |
> | E1-Skywork-OR1-Math-7B  | 11768 | 69.6 | 1381 | 16.9 | 1841 | 21.3 | 2799  | 26.0 | 3742 | 32.9 |
>
> These results further confirm that the proposed *Elastic Reasoning* framework is not specific to a single model but is broadly applicable.
>
> [1] https://huggingface.co/Skywork/Skywork-OR1-Math-7B

---

### Author Response · Authors · 2025-12-01
**General Comments**

We appreciate the AC and reviewers for their careful review and valuable insights throughout the rebuttal phase. We have revised the paper accordingly and updated the PDF on OpenReview. All modifications in the main article are marked. Below is a summary of the revisions.

1. Clarified Section 3.2.3 with a clearer comparison to vanilla GRPO.
   **(Reviewer 8Dnp)**
2. Clarified the meaning of the red squares in the caption of Figure 2.
   **(Reviewer 8Dnp)**
3. Updated captions for Figures 1, 4, 5, and 6 to clearly explain the x-axis meaning.
   **(Reviewer apFU)**

4. Added model-variant experiments on Skywork-OR1-Math-7B in Appendix I.
   **(Reviewer apFU)**

5. Included detailed GRPO training setup in Appendix B.
   **(Reviewer apFU)**

6. Added discussion on why E1-Math-1.5B slightly underperforms the original model in Appendix O.1.
   **(Reviewer apFU)**

7. Added experiments varying the solution budget, and zero-thinking results in Appendices J and K.
   **(Reviewer apFU and Reviewer 9D8U)**

8. Reported results with standard deviation in Appendix L.
   **(Reviewer apFU)**

9. Added budget-aligned comparison results in Appendix M.
   **(Reviewer 9D8U)**

10. Explained the choice of the 1K solution budget in Appendix O.2.
    **(Reviewer 9D8U)**

11. Included comparisons of varying thinking vs. solution lengths under equal total budgets in Appendix N.
    **(Reviewer 9D8U)**

---

### Meta-Review · Area_Chair_8pSb · 2025-12-20

**Summary:**

The paper proposes Elastic Reasoning, a novel framework that separates thinking and solution budgets to enable scalable and controllable CoT inference. Reviewers agreed that this method is simple and effective, showing strong empirical results on math and coding benchmarks. Moreover, the authors provided a comprehensive rebuttal, addressing concerns by adding statistical rigor, demonstrating generalization to additional models, and clarifying comparisons with baselines. Therefore, I recommend acceptance.

**Reviewer Concerns:**

Addressed:
1. The method was only tested on DeepScaleR/DeepCoder. It was unclear if it worked on other architectures. The authors added experiments using Skywork-OR1-Math-7B.
2. The original figures lacked error bars, making it hard to judge significance. The authors provided tables with mean ± standard deviation.
3. Confusion regarding how the model knows the budget and how this differs from S1. Authors clarified that no budget tokens are input to the model; truncation is handled externally. They distinguished their method from S1 by explaining that they control both thinking and solution budgets, whereas S1 only controls thinking via prompts.

Outstanding:
I do not see very strong outstanding concerns remaining.

**Reviewer Scores:**

Reviewer apFU raised score from 2 to 4.
Other reviewers may keep their score unchanged, because their scores are already positive.

---

### Decision · Program_Chairs · 2026-01-26

Accept (Poster)